# WIMLE: Uncertainty-Aware World Models with IMLE for Sample-Efficient Continuous Control

**Mehran Aghabozorgi, Alireza Moazeni, Yanshu Zhang, Ke Li**
APEX Lab, School of Computing Science
Simon Fraser University
{mehran_aghabozorgi,sam62,yanshu_zhang,keli}@sfu.ca

## Abstract

Model-based reinforcement learning promises strong sample efficiency but often underperforms in practice due to compounding model error, unimodal world models that average over multi-modal dynamics, and overconfident predictions that bias learning. We introduce WIMLE, a model-based method that extends Implicit Maximum Likelihood Estimation (IMLE) to the model-based RL framework to learn stochastic, multi-modal world models without iterative sampling and to estimate predictive uncertainty via ensembles and latent sampling. During training, WIMLE weights each synthetic transition by its predicted confidence, preserving useful model rollouts while attenuating bias from uncertain predictions and enabling stable learning. Across 40 continuous-control tasks spanning DeepMind Control, MyoSuite, and HumanoidBench, WIMLE achieves superior sample efficiency and competitive or better asymptotic performance than strong model-free and model-based baselines. Notably, on the challenging Humanoid-run task, WIMLE improves sample efficiency by over 50% relative to the strongest competitor, and on HumanoidBench it solves 8 of 14 tasks (versus 4 for BRO and 5 for SimbaV2). These results highlight the value of IMLE-based multi-modality and uncertainty-aware weighting for stable model-based RL.

## 1 Introduction

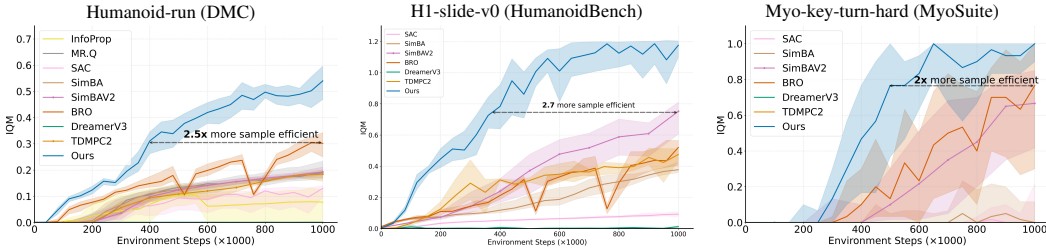

Figure 1: Sample efficiency on challenging tasks from each benchmark suite. WIMLE achieves superior sample efficiency and asymptotic performance over strong model-free and model-based baselines. Y-axes show interquartile mean. Shaded areas indicate 95% confidence intervals.

Reinforcement learning has become a powerful framework for solving complex decision-making problems across diverse domains such as autonomous control (Kiumarsi et al., 2018), strategic game playing (Hosu & Rebedea, 2016), and natural language processing (Lambert, 2025; Cetina et al., 2021). However, a significant challenge in RL is the need for a substantial number of interactions with the environment to learn a good policy. Without a simulator, learning requires real-world trials, which are costly, slow, and risky (Chen et al., 2022; Weng et al., 2023; Hessel et al., 2018; Schulman et al., 2017b).

Model-based RL (MBRL) methods aim to address this sample efficiency issue by first learning a parametric world model from collected environment interactions, then leveraging this learned

model to reduce real environment samples and accelerate policy learning (Hafner et al., 2020; 2021; 2023; Ye et al., 2021; Laskin et al., 2020). Common uses of the learned world model include (1) generating synthetic rollouts that augment training data for policy learning (Janner et al., 2019; Ha & Schmidhuber, 2018; Hafner et al., 2020; Clavera et al., 2020) and (2) planning by simulating future trajectories to guide action selection (Zhu et al., 2023; Frauenknecht et al., 2025; Janner et al., 2019; Lowrey et al., 2019; Hafner et al., 2019; Argenson & Dulac-Arnold, 2021). In this work, we focus on the former.

Historically, MBRL has struggled to surpass strong model-free baselines, largely because compounding rollout errors bias training and mislead the policy (Janner et al., 2019; Xiao et al., 2019; Talvitie, 2017; Frauenknecht et al., 2025; Venkatraman et al., 2015; Asadi et al., 2018b;a). We attribute this to two key issues: (1) standard predictive models struggle when the same state–action pair yields different, conflicting supervision due to partial observability, contact-rich dynamics, or inherent stochasticity (Kurutach et al., 2018); and (2) a lack of uncertainty awareness in model predictions (Frauenknecht et al., 2025), which leads to overconfidence in regions with complex dynamics or limited data. Despite attempts to address these issues (Janner et al., 2019; Zhu et al., 2023; Frauenknecht et al., 2025; Somalwar et al., 2025; Hansen et al., 2022), MBRL methods have yet to consistently outperform strong model-free baselines in practice (Nauman et al., 2024; Lee et al., 2025b).

To address these issues, we propose WIMLE (**W**orld models with **IMLE**)—an uncertainty-aware model-based RL approach. We integrate IMLE (Li & Malik, 2018), a mode-covering generative model with demonstrated success in low-data regimes (Aghabozorgi et al., 2023; Vashist et al., 2024), into the MBRL framework. This allows us to learn world models that handle different, conflicting supervision and from which we extract predictive uncertainty estimates. We incorporate these uncertainty estimates into the RL objective to prevent overconfident predictions from biasing learning. To the best of our knowledge, this is the first work to extend IMLE for uncertainty-aware world models in MBRL.

We evaluate WIMLE on $40$ tasks across DMC, HumanoidBench, and MyoSuite. WIMLE delivers considerable gains in sample efficiency and asymptotic performance over strong model-free and model-based baselines. Notably, on the notoriously challenging Humanoid-run task, WIMLE improves the sample efficiency of the most competitive method by over $50\%$. On HumanoidBench, WIMLE successfully solves $8$ of $14$ tasks, compared to $4$ for BRO and $5$ for SimbaV2 (Figure 10). Across suites, Figure 1 shows one example task per benchmark, each showing more than $50\%$ sample-efficiency improvement for WIMLE over the strongest competing method.

## 2 PRELIMINARIES

### 2.1 RL

We consider an infinite-horizon discounted Markov decision process (MDP) $(\mathcal{S}, \mathcal{A}, P, r, \gamma)$ (Bellman, 1957) with initial state distribution $\rho_0$. At time $t$, the agent observes $s_t \in \mathcal{S}$, selects $a_t \sim \pi_\phi(a \mid s_t)$, receives reward $r_t = r(s_t, a_t)$, and the environment transitions as $s_{t+1} \sim P(\cdot \mid s_t, a_t)$. The objective is to learn a policy that maximizes the expected discounted return

$$J(\pi_\phi) \;=\; \mathbb{E}_{\tau \sim (\rho_0, P, \pi_\phi)}\left[ \sum_{t=0}^{\infty} \gamma^t \, r(s_t, a_t) \right]. \tag{1}$$

In the continuous-control settings considered here, states and actions are real-valued. The discount factor satisfies $\gamma \in (0, 1)$. We denote the action-value function under policy $\pi$ by $Q^\pi(s, a)$:

$$Q^\pi(s, a) = \mathbb{E}_{\tau \sim (\pi, P)}\left[ \sum_{t=0}^{\infty} \gamma^t r(s_t, a_t) \mid s_0 = s, a_0 = a \right]. \tag{2}$$

### 2.2 MODEL-BASED RL

In model-based RL, we learn a parametric world model with parameters $\theta$ that approximates the unknown environment transition dynamics $P(s_{t+1}, r_t \mid s_t, a_t)$ through a learned conditional distribution

$$\hat{p}_\theta\big(s_{t+1}, r_t \mid s_t, a_t\big) \tag{3}$$

trained from limited environment interactions. A model rollout (prediction) of horizon $H$ under a policy $\pi_\phi$ from $s_0$ is the sequence $\hat{\tau} = (s_0, a_0, r_0, s_1, \ldots, s_H)$ generated by

$$a_t \sim \pi_\phi(\cdot \mid s_t), \quad (s_{t+1}, r_t) \sim \hat{p}_\theta(\cdot \mid s_t, a_t). \tag{4}$$

Such rollouts are commonly used for planning or to provide synthetic transitions for RL training (Janner et al., 2019; Zhu et al., 2023).

### 2.3 IMPLICIT MAXIMUM LIKELIHOOD ESTIMATION

Implicit Maximum Likelihood Estimation (IMLE) (Li & Malik, 2018) learns a latent-variable generator $g_\theta(z)$ that maps noise $z \sim \mathcal{N}(0, I)$ to data space. Given data $\{x_i\}_{i=1}^N$, the IMLE objective is:

$$\theta^\star = \arg\min_\theta \ \mathbb{E}_{\{z_j\}_{j=1}^m} \sum_{i=1}^N \min_{1 \le j \le m} \left\| g_\theta(z_j) - x_i \right\|^2. \tag{5}$$

In practice, given current parameters $\theta$, we realize this objective by drawing a pool of candidate latents $\{z_j\}_{j=1}^m$ i.i.d. from $\mathcal{N}(0, I)$ per data point and selecting the nearest generated sample; this step is gradient-free and fully parallelizable,

$$z_i^\star = \arg\min_{1 \le j \le m} \left\| g_\theta(z_j) - x_i \right\|^2, \tag{6}$$

and minimizing the resulting empirical loss using stochastic gradient descent.

$$\theta \leftarrow \theta - \eta \, \nabla_\theta \frac{1}{|B|} \sum_{i \in B} \left\| g_\theta(z_i^\star) - x_i \right\|^2. \tag{7}$$

Optimizing Eq. equation 5 yields maximum likelihood estimation (MLE) of $\theta$ and ensures mode coverage (Aghabozorgi et al., 2023): each data point is represented by at least one generated sample. In practice, IMLE is sample efficient and effective for modeling multi-modal distributions. Conditional IMLE (Li et al., 2020) $g_\theta(c, z)$ models multi-modal conditional distributions; we adopt this form in WIMLE. For a more detailed discussion of IMLE, we refer readers to (Li & Malik, 2018; Aghabozorgi et al., 2023).

## 3 WIMLE

WIMLE addresses the key limitations of traditional MBRL through three main components: (1) IMLE-trained stochastic world models that capture complex multi-modal transition dynamics, (2) predictive uncertainty estimation that reflects the model's confidence in its predictions, and (3) uncertainty-weighted learning that scales the influence of synthetic data based on model confidence. We detail each component below.

### 3.1 IMLE WORLD MODEL

Recent MBRL approaches span autoregressive sequence models, latent-variable generators, diffusion models, and planning-centric objectives (Ha & Schmidhuber, 2018; Hafner et al., 2019; 2020; 2021; Robine et al., 2023; Micheli et al., 2023; Zhang et al., 2023; Hansen et al., 2024; Huang et al., 2024). Diffusion models are effective but rely on iterative sampling, which limits their usage in the online RL setting where rollout throughput is critical (Huang et al., 2024; Karras et al., 2022). Despite progress, these methods often require substantial data and still struggle to consistently surpass strong model-free baselines (Nauman et al., 2024; Lee et al., 2025b). Moreover, simple and sample-efficient unimodal Gaussian world models underfit inherently multi-modal, complex dynamics in partially observable or contact-rich settings, exacerbating model bias and compounding errors (Janner et al., 2019; Zhu et al., 2023).

On the other hand, we leverage IMLE to learn transitions, a one-step generative method that—unlike diffusion models—avoids iterative sampling and enables fast online rollouts. In practice, IMLE yields strong rollout throughput; Figure 2 reports wall-clock time among model-based methods. We represent the world model as a conditional stochastic generator $g_\theta$ that maps a state–action pair and latent noise to the next outcome:

$$(\tilde{s}_{t+1}, \tilde{r}_t) = g_\theta(s_t, a_t, z), \quad z \sim \mathcal{N}(0, I). \tag{8}$$

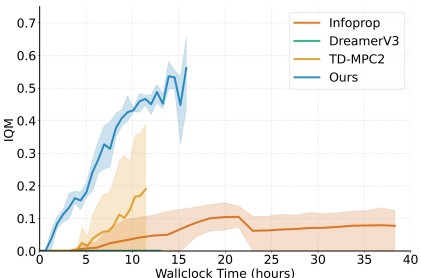

Figure 2: Wall-clock comparison among model-based methods (3 seeds) on a single NVIDIA L40S GPU for the humanoid-run task. Y-axis shows interquartile mean; shaded areas indicate 95% confidence intervals.

Here, the latent variable $z$ induces a distribution over next outcomes for the same state–action pair, capturing inherent stochasticity and multi-modality in the dynamics.

**IMLE Training Procedure.** Given a dataset of transitions $\{(s_t, a_t, r_t, s_{t+1})\}_{i=1}^{N}$, we form targets $y_i = [r_t, s_{t+1}]$ and train $g_\theta$ using the IMLE objective. The training proceeds in two alternating steps:

*Assignment Step:* For each data point $y_i$, we sample $m$ candidate latents $\{z_j\}_{j=1}^{m}$ and assign the nearest candidate that minimizes the prediction error:

$$z_i^\star \;=\; \arg \min_{1 \leq j \leq m} \; \left\| g_\theta(s_i, a_i, z_j) - y_i \right\|^2. \tag{9}$$

This assignment step is computationally efficient as it requires no gradient computation, is fully parallelizable across data points, and typically uses small values of $m$ (e.g., 5-10) in conditional IMLE settings.

*Update Step:* We then perform gradient descent on the empirical loss using the assigned latents:

$$\theta \leftarrow \theta - \eta \, \nabla_\theta \, \frac{1}{|B|} \sum_{i \in B} \left\| g_\theta(s_i, a_i, z_i^\star) - y_i \right\|^2, \tag{10}$$

where $B$ is a minibatch of indices and $\eta > 0$ is the learning rate.

This procedure ensures *mode coverage* by matching each data point to at least one generated sample, avoiding collapse to a single mean prediction. In contrast, a standard Gaussian regression model trained with least squares (Janner et al., 2019) predicts the conditional mean; in multi-modal settings this falls between modes and produces averaged, often implausible next states—known as regression to the mean (Bruna et al., 2016)—that compound over rollouts. IMLE's per-sample latent assignment avoids this averaging and yields sharper, mode-consistent predictions (Aghabozorgi et al., 2023; Vashist et al., 2024).

**Inference and Rollouts.** After training, we generate rollouts following the procedure described in Section 2.2. Multi-step rollouts of horizon $H$ are generated by initializing from a real state $s_0$ and iteratively applying: $a_t \sim \pi_\phi(\cdot|s_t), z_t \sim \mathcal{N}(0, I), (s_{t+1}, r_t) = g_\theta(s_t, a_t, z_t)$ for $t = 0, \ldots, H-1$.

## 3.2 Uncertainty Estimation

Reliable uncertainty estimation is crucial for deciding when to trust model predictions. We therefore compute a predictive uncertainty for each synthetic transition and use it to reweight the RL objective. Each transition's contribution is scaled by its estimated confidence. This preserves useful rollouts and reduces bias from uncertain predictions without changing the underlying algorithm. Alternative integrations exist. For example, Infoprop (Frauenknecht et al., 2025) computes an information-theoretic corruption measure and uses it to truncate rollouts during generation. In contrast, we integrate uncertainty directly into the learning objective via confidence weights.

We use a single predictive uncertainty measure $\sigma(s, a)$ that reflects the model's confidence in its next-step prediction. Concretely, we maintain an ensemble of $K$ IMLE world models (see Section 3.6.1) and, for each model, draw $m$ latent samples, yielding predictions:

$$\{g_{\theta_k}(s, a, z_j)\}_{k=1..K, \, j=1..m} \tag{11}$$

We define $\sigma(s, a)$ as the standard deviation across these predictions—a direct measure of model agreement (see Algorithm 1, lines 13–14):

$$\sigma(s, a) = \text{std}_{k,j}\left[g_{\theta_k}(s, a, z_j)\right] \tag{12}$$

In practice, we compute per-dimension standard deviations for the predicted reward and next state and average them to obtain a single scalar for the transition. $\sigma(s, a)$ decreases when models agree and increases when predictive uncertainty is high (e.g., limited data or complex dynamics). By the law of total variance we can decompose $\sigma^2(s, a)$ into epistemic and aleatoric components,

$$\sigma^2(s,a) = \underbrace{\mathrm{Var}_k\left[\mathbb{E}_z\left[g_{\theta_k}(s,a,z)\right]\right]}_{\text{ensemble / epistemic}} + \underbrace{\mathbb{E}_k\left[\mathrm{Var}_z\left[g_{\theta_k}(s,a,z)\right]\right]}_{\text{latent / aleatoric}}, \tag{13}$$

Following recent insights on uncertainty from Smith et al. (2024), this total predictive variance is the Bayes risk of acting under a squared-error loss.

### 3.3 UNCERTAINTY-WEIGHTED LEARNING

Having defined a single predictive uncertainty $\sigma(s, a)$, we now describe how it enters learning. The key idea is simple: weight each synthetic transition by the model's confidence so reliable predictions contribute more and uncertain ones less. Because uncertainty typically grows with rollout horizon due to error accumulation, later steps in the rollout receive smaller weights, naturally down-weighting distant, noisier predictions.

To choose these weights, we invoke a standard heteroscedastic regression result (formalized in Section 3.4): when estimating a scalar quantity from independent noisy observations with different noise variances, the unique minimum-variance linear unbiased estimator weights each observation inversely to its variance. We use this principle as guidance and map each predictive standard deviation $\sigma(s, a)$ to a bounded weight

$$w(s,a) = \frac{1}{\sigma(s,a) + 1}, \tag{14}$$

which preserves the inverse-variance ordering and keeps $w \in (0, 1]$, avoiding exploding gradients.

Our analysis suggests the idealized scaling $w(s, a) \propto 1/\sigma^2(s, a)$. In practice, we use the bounded proxy $w(s, a) = 1/(1 + \sigma(s, a))$, which we find empirically to be more stable and to work better.

During rollout generation, we compute per-transition weights $w_i = w(s_i, a_i)$ for each synthetic transition $(s_i, a_i, r_i, s_i')$ using the predictive uncertainty defined above. We incorporate these weights into the RL objective by modifying the temporal difference (TD) loss:

$$\mathcal{L}_{\text{critic}} = \mathbb{E}_{(s_i,a_i,r_i,s_i')\sim\mathcal{D}}\left[w_i \cdot \delta_i^2\right], \tag{15}$$

where $\delta_i = r_i + \gamma Q_\phi(s_{i+1}, a_{i+1}) - Q_\phi(s_i, a_i)$ is the TD error for transition $i$, with $a_{i+1} \sim \pi_\phi(\cdot \mid s_{i+1})$, $Q_\phi$ is a parameterized Q-function, and $w_i$ is the corresponding uncertainty weight. For real environment data, we simply use $w_i = 1$.

This approach lets us leverage synthetic rollouts while keeping TD updates well-conditioned: high-variance, uncertain transitions have a smaller impact on the stochastic gradients, while the Bellman fixed point remains unchanged, so we tame noisy (including purely stochastic) updates without discarding or biasing them.

### 3.4 THEORETICAL ANALYSIS

To make the effect of our weighting more concrete, we provide a theoretical analysis of its properties. First, we establish that multiplying each squared Bellman error $(y - Q(s, a))^2$ by a positive weight $w(s, a)$ preserves the Bellman fixed point. Second, in a tractable setting where the critic is linear in a feature representation, we demonstrate that choosing weights inversely proportional to the (total) target noise variance minimizes the variance of the learned parameters, thereby improving convergence rate and sample efficiency. The lemmas below formalize these results: the first shows that any strictly positive reweighting leaves the Bellman target $Q^\star$ unchanged, and the second shows that, in the linear-critic regime, inverse-variance weighting $w_i \propto 1/\sigma_i^2$ yields the minimum-covariance linear unbiased estimator. Full proofs are provided in Appendix B.

**Lemma (Positive weights preserve the Bellman target).** Let $y = r + \gamma V(s')$ denote the one-step Bellman target for a value function $V$ with conditional mean $\mu(s, a) = \mathbb{E}[y \mid s, a]$. Given a strictly positive weight function $w(s, a) > 0$, the corresponding population weighted Bellman squared loss is

$$\mathcal{L}_w(Q) = \mathbb{E}_{(s,a)\sim d^\pi}\,\mathbb{E}_{(r,s')\sim P(\cdot|s,a)}\left[w(s,a)\left(y - Q(s,a)\right)^2\right]. \tag{16}$$

Then the unique minimizer of $\mathcal{L}_w(Q)$ over all action-value functions $Q$ is $Q^\star(s,a) = \mu(s,a)$ for all $(s,a)$, i.e., any strictly positive reweighting leaves the Bellman fixed point unchanged and only re-emphasizes different regions of $(s,a)$. A full proof is given in Appendix B.

**Lemma (Linear critics, inverse-variance weighting).** Consider the setting where (1) the action-value function is linear in some feature representation $\phi(s,a) \in \mathbb{R}^d$, i.e., $Q_\theta(s,a) = \phi(s,a)^\top\theta$ for parameters $\theta \in \mathbb{R}^d$, and (2) for a batch of $m$ transitions $x_i = (s_i, a_i)$ the TD targets satisfy $y_i = \mu(x_i) + \varepsilon_i$ with $\mathbb{E}[\varepsilon_i \mid x_i] = 0$, $\mathrm{Var}(\varepsilon_i \mid x_i) = \sigma_i^2$, and independent noise terms $\varepsilon_i$. Among all linear unbiased estimators of $\theta$, the inverse-variance choice $w_i \propto 1/\sigma_i^2$ yields the minimum covariance matrix for $\hat{\theta}$. A proof follows the classical Gauss–Markov theorem and is provided in Appendix B.

*Proof sketch.* Stacking targets into a vector $y$ and features into a design matrix $\Phi$, minimizing $\sum_i w_i(y_i - Q_\theta(x_i))^2$ yields the weighted least-squares estimator $\hat{\theta}_w = (\Phi^\top W \Phi)^{-1}\Phi^\top W y$ with $W = \mathrm{diag}(w_i)$. Choosing $W$ proportional to the inverse noise covariance $\Sigma^{-1} = \mathrm{diag}(1/\sigma_i^2)$ yields an estimator whose covariance $\mathrm{Cov}(\hat{\theta}_w)$ is minimal among all linear unbiased estimators by the Gauss–Markov theorem. We refer to Appendix B for a full derivation.

**Implications for WIMLE.** Here $\sigma_i^2$ is the total predictive variance at $(s_i, a_i)$, which can include both epistemic and aleatoric components (Eq. 13). The lemma above shows that, in the linear-critic regime, weighting each transition inversely to this total variance minimizes the covariance of the learned parameters, independently of the source of the noise. Lower parameter variance means fewer samples are needed to reach a given accuracy, i.e., inverse-variance weighting is provably more sample-efficient in this regime. Combined with the Bellman fixed-point lemma above (proved in Appendix B), this means our choice of $w(s,a)$ shrinks update variance without ever changing the Bellman solution—even in the limit of a perfect world model where all uncertainty (and thus down-weighted noise) is purely aleatoric.

## 3.5 ALGORITHM

Algorithm 1 presents the overall WIMLE procedure. For a more complete implementation of the algorithm, including training frequencies and hyperparameters, see Algorithm 3 in Appendix A.

---

**Algorithm 1** WIMLE: World Models with Implicit Maximum Likelihood Estimation

---

1: **Input:** Rollout horizon $H$, ensemble size $K$, number of rollouts $M$, number of latent codes $m$
2: {Red text indicates steps that fundamentally differ from MBPO.}
3: initialize ensemble world models $\{g_{\theta_k}\}_{k=1}^K$, environment and model datasets $\mathcal{D}_{\mathrm{env}}, \mathcal{D}_{\mathrm{model}}$
4: **for** environment steps **do**
5:     Collect environment transitions using $\pi_\phi$; add to $\mathcal{D}_{\mathrm{env}}$
6:     **// IMLE World Model Training**
7:     Train ensemble $\{g_{\theta_k}\}_{k=1}^K$ in parallel on bootstrap samples of $\mathcal{D}_{\mathrm{env}}$ using IMLE (Eqs. 9, 10)
8:     **for** $M$ model rollouts **do**
9:         Sample starting state $s_0$ from $\mathcal{D}_{\mathrm{env}}$
10:         **for** $t = 0$ to $H - 1$ **do**
11:             $a_t \sim \pi_\phi(\cdot|s_t)$
12:             Sample $m$ latents $\{z_j\}_{j=1}^m \sim \mathcal{N}(0, I)$
13:             Generate predictions $\{(\tilde{s}_{t+1}, \tilde{r}_t)_{k,j}\}_{k=1,j=1}^{K,m} = \{g_{\theta_k}(s_t, a_t, z_j)\}_{k=1,j=1}^{K,m}$ from all ensemble members
14:             Compute predictive uncertainty: $\sigma_t = \mathrm{std}_{k,j}[g_{\theta_k}(s_t, a_t, z_j)]$ {aggregated over ensembles and latents}
15:             Set weight $w_t = 1/(\sigma_t + 1)$
16:             Add weighted transition $(s_t, a_t, r_t, s_{t+1}, w_t)$ to $\mathcal{D}_{\mathrm{model}}$
17:     **// Uncertainty-Weighted Policy Learning**
18:     Sample batch from $\mathcal{D}_{\mathrm{env}} \cup \mathcal{D}_{\mathrm{model}}$ (real data has $w = 1$)
19:     Update policy using weighted RL objective:
20:         $\mathcal{L} = \mathbb{E}_{(s,a,r,s',w)\sim\mathrm{batch}}[w \cdot \ell_{\mathrm{RL}}(s,a,r,s')]$

---

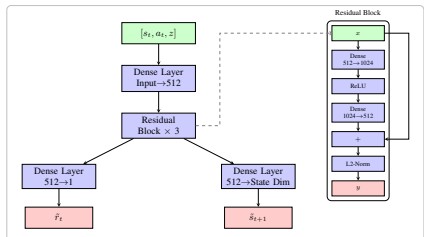

Figure 3: WIMLE world model architecture.

The algorithm maintains the underlying RL method as a black box through the weighted loss function $\mathcal{L}$, where $\ell_{\mathrm{RL}}$ represents any standard RL objective (e.g., TD error for critics, policy gradient for actors). The key insight is that uncertainty weights $w_t$ automatically scale the contribution of each synthetic transition—high-confidence predictions receive higher weights while uncertain predictions contribute proportionally less.

### 3.6 DESIGN CHOICES

#### 3.6.1 TRAINING

**Model Rollouts.** We experiment with synthetic rollouts using horizons up to $H = 8$. All rollouts are initialized from real environment states sampled uniformly from the environment dataset $\mathcal{D}_{\mathrm{env}}$, following standard practice in model-based RL to ensure rollouts start from the data distribution. We select task-specific rollout horizons through empirical experimentation as described in Appendix D.1.

**RL Training.** We use Soft Actor-Critic (SAC) (Haarnoja et al., 2018) with distributional Q-learning (Bellemare et al., 2017) as our underlying RL algorithm. Following recent work that has demonstrated the effectiveness of distributional RL for continuous control (Nauman et al., 2024; Lee et al., 2025b; Dabney et al., 2018a), we specifically adapt quantile Q-learning (Dabney et al., 2018b; Nauman et al., 2024).

**Ensemble Training.** We train an ensemble of $K = 7$ IMLE world models in parallel to improve predictive uncertainty estimation and calibration (Section 3.3). Each ensemble member is initialized with different random parameters and trained on bootstrap samples of the environment data. The parallel training of ensemble members is computationally efficient and scales well with available compute resources, enabling reliable predictive uncertainty without significant computational overhead.

#### 3.6.2 ARCHITECTURE

Figure 3 illustrates the WIMLE world model architecture. The network takes as input state $s_t$, action $a_t$, and latent variable $z$, followed by a dense layer that maps to a 512-dimensional hidden representation. The core of the architecture consists of three residual blocks, each containing dense layers with ReLU activations and L2 normalization. Following recent findings by Lee et al. (2025b), we employ L2 normalization within the residual blocks, which has been shown to improve stability and performance in RL settings. The network outputs separate predictions for rewards and next states through dedicated dense heads.

## 4 EXPERIMENTS

We evaluate WIMLE across diverse continuous-control benchmarks—DeepMind Control Suite (including Dog and Humanoid), MyoSuite, and HumanoidBench (Tassa et al., 2018; Caggiano et al., 2022; Sferrazza et al., 2024). Across 40 tasks spanning locomotion and dexterous manipulation with high-dimensional state/action spaces and sparse rewards, we compare against strong model-free and model-based methods, including MR.Q, PPO, SAC, Simba, SimbaV2, BRO, TD-MPC2, and DreamerV3 (Fujimoto et al., 2025; Schulman et al., 2017a; Haarnoja et al., 2018; Lee et al., 2025a;b; Nauman et al., 2024; Hansen et al., 2024; Hafner et al., 2023), and present per-benchmark results. Through our experiments, we aim to answer: (i) How does WIMLE compare to strong model-free and model-based methods? (ii) How does IMLE-based multi-modality in the world

model affect results compared to standard unimodal Gaussian models? (iii) How do uncertainty estimates evolve during training, and how do they affect performance?

## 4.1 EXPERIMENTAL SETUP

All experiments are run for 1M environment steps with 10 random seeds unless otherwise specified. We report the interquartile mean (IQM) and 95% confidence intervals computed with RLiable (Agarwal et al., 2021), using stratified bootstrap across tasks and seeds. Following BRO and SimbaV2 (Nauman et al., 2024; Lee et al., 2025b), we aggregate normalized scores per BRO/SimbaV2 protocol (DMC [0,1], MyoSuite success, HumanoidBench success-normalized). Where official baseline results are available, we report the authors' numbers; otherwise, we run public implementations with their recommended settings. We provide full details about the experimental setup, hyperparameters, and baselines in Section D and E of the appendix.

## 4.2 COMPARISON TO BASELINES

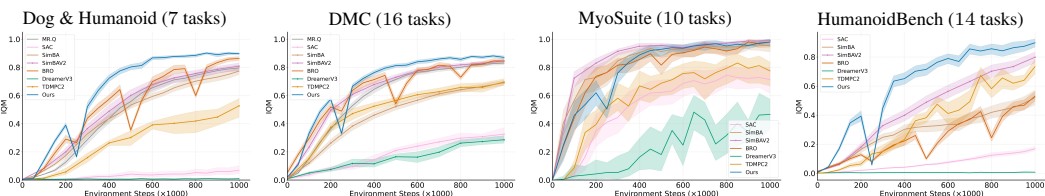

Figure 4: Aggregate results across benchmarks. WIMLE outperforms strong model-free and model-based baselines overall. Gains are most pronounced on the challenging Dog & Humanoid subset, where it achieves superior sample efficiency and asymptotic performance. On MyoSuite, it performs asymptotically on par with strong baselines that are already near the maximum score (1.0), and on HumanoidBench it significantly outperforms the baselines, solving $8/14$ tasks versus BRO 4 and SimbaV2 5. Y-axes show interquartile mean; shaded areas denote 95% confidence intervals.

We summarize aggregate performance across benchmarks in Figure 4 and provide detailed per-task results in Section C of the appendix. WIMLE consistently leads among strong model-free and model-based methods on Dog & Humanoid, the full DMC suite, and HumanoidBench, while performing asymptotically on par with strong MyoSuite baselines that are already close to the maximum score (1.0). Notably, gains are largest on the high-dimensional and challenging Dog & Humanoid tasks (Dog: $|\mathcal{S}|$=223, $|\mathcal{A}|$=38; Humanoid: $|\mathcal{S}|$=67, $|\mathcal{A}|$=24). On HumanoidBench, WIMLE significantly outperforms baselines, solving 8 of 14 tasks versus BRO 4 and SimbaV2 5 (Figure 10). We summarize performance across timesteps in Section E.1, where WIMLE performs best or competitively across most evaluations. We attribute these improvements to IMLE-driven multi-modality in the world model and uncertainty-weighted learning that scales the influence of synthetic rollouts by model confidence, mitigating bias from overconfident predictions while preserving useful signal, which we discuss in more detail in the next section. Per-task performance is reported in Figures 7, 8, 9 and 10.

## 4.3 METHOD ANALYSIS

We analyze how uncertainty-aware weighting and multi-modal dynamics modeling impact performance and how predictive uncertainty evolves during training.

**Effect of uncertainty-aware weighting** Figure 5 (Left) compares WIMLE with uncertainty-aware weighting to an unweighted variant that is *identical in every respect except that all per-transition weights are fixed to* $w_i$=1.0. The unweighted curve lags and can even underperform a strong model-free baseline early on, indicating that ignoring predictive uncertainty significantly biases learning and hinders performance. Figure 5 (Right) studies rollout sensitivity. Increasing the model rollout horizon from $H$=1 to $H$=4 to $H$=6 improves performance, and extending to $H$=8 maintains performance rather than showing the severe degradation typically observed in model-based methods when increasing rollout length (Janner et al., 2019). This improved stability at longer horizons demonstrates that uncertainty-aware weighting reduces the model bias typically introduced by longer horizon errors, enabling us to leverage longer synthetic rollouts without considerable performance degradation.

**Impact of IMLE-based multi-modality**   Figure 6 (Left) contrasts WIMLE (IMLE world model) with an otherwise identical unimodal Gaussian world model (MBPO-style; (Janner et al., 2019)), with both variants using uncertainty-aware weighting. The IMLE variant significantly outperforms the Gaussian, underscoring the value of modeling multi-modal transition dynamics for uncertainty estimation in complex, contact-rich control. Figure 6 (Right) shows how weights evolve. During a brief warm-up with limited environment samples and training, both models are uncalibrated and weights can appear transiently high; as data accumulates and the estimators calibrate, weights drop to reflect high uncertainty and low confidence. As training progresses and more data are collected, IMLE's weights increase to reflect higher confidence in the predictions, whereas the Gaussian's remain relatively flat, indicating limited calibration. Together, these results show that multi-modal modeling improves both performance and the quality of uncertainty estimates, reducing the risk of overconfident, biased predictions misleading the policy.

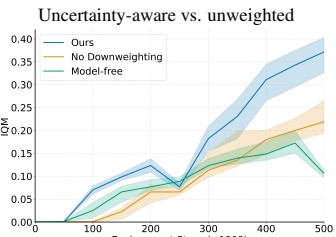 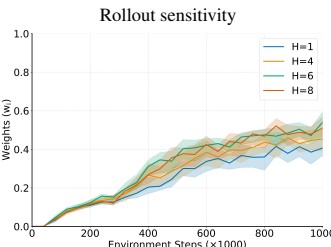

Figure 5: Uncertainty-aware weighting reduces model bias and enables stable training at longer horizons on Humanoid-run. **Left:** Uncertainty-aware WIMLE compared to an unweighted variant that is identical except all per-transition weights are fixed to $w_i = 1.0$ and a model-free variant that is identical except that it does not use the model; the unweighted curve lags and can even underperform the model-free variant early on, indicating that ignoring uncertainty will bias learning and hinder performance. **Right:** Rollout ablation ($H = 1, 4, 6, 8$) for WIMLE: increasing the model rollout horizon from $H=1$ to $H=4$ to $H=6$ improves performance, and extending to $H=8$ does not substantially degrade performance, suggesting that uncertainty-aware weighting mitigates harm from error accumulation at longer horizons. All variants use the same SAC backbone and distributional critics; only the ablated components differ. All plots are on DMC's Humanoid-run task with 5 seeds.

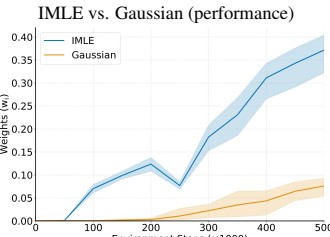 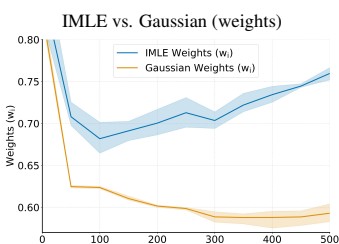

Figure 6: Multi-modality strengthens model-based learning. **Left:** WIMLE (IMLE world model) versus an otherwise identical unimodal Gaussian world model (MBPO-style; (Janner et al., 2019); both use uncertainty-aware weighting): the IMLE variant significantly outperforms the Gaussian, highlighting the value of multi-modal modeling and IMLE's efficacy. **Right:** Weight dynamics: After a brief warm-up phase, IMLE's weights are lower when uncertainty is high and increase as training progresses and more data are collected, reflecting growing model confidence; the unimodal Gaussian fails to capture this evolution, yielding relatively flat weights over time. All plots are on Humanoid-run.

## 5 RELATED WORK

**Model-free RL.**   Foundational model-free methods such as PPO (Schulman et al., 2017a) and SAC (Haarnoja et al., 2018) remain strong references for continuous control. Recent advances focus on scaling and regularization: BRO (Nauman et al., 2024) scales critic networks to 5M parameters with strong regularization and optimistic exploration, achieving state-of-the-art performance. Simba (Lee et al., 2025a) introduces an architecture that embeds simplicity bias through running statistics

normalization, residual feedforward blocks, and layer normalization, enabling effective parameter scaling; SimbaV2 (Lee et al., 2025b) further constrains feature and weight norms via hyperspherical normalization. Contemporary work like MR.Q (Fujimoto et al., 2025) explores improved value estimation for better sample efficiency. Collectively, these methods provide strong model-free baselines.

**Model-based RL.** Model-based RL methods learn world models to improve sample efficiency via synthetic rollouts and planning. DreamerV3 (Hafner et al., 2023) learns a latent world model and achieves strong performance in continuous control with large-scale training. MBPO (Janner et al., 2019) uses short model-generated rollouts branched from real data to avoid model exploitation while maintaining sample efficiency. TD-MPC2 (Hansen et al., 2024) learns implicit world models through joint-embedding prediction and performs local trajectory optimization in latent space for scalable multi-task learning. STORM (Zhang et al., 2023) combines Transformer-based sequence modeling with categorical VAEs for efficient world model learning in visual domains. Diffusion-based world models generate trajectories via iterative denoising and incur high inference cost, which hinders online RL (Janner et al., 2022; Ajay et al., 2023; He et al., 2023). Despite these algorithmic advances, model-based methods have struggled to consistently surpass recent model-free approaches like BRO (Nauman et al., 2024) and SimbaV2 (Lee et al., 2025b).

**Model Bias.** Model bias and error accumulation remain fundamental challenges in MBRL. Trajectory models (Asadi et al., 2019; Lambert et al., 2021) address the compounding-error problem by learning multi-step models that directly predict outcomes of action sequences, avoiding the accumulation of one-step prediction errors. Self-correcting models (Talvitie, 2017) train models to correct themselves when producing errors. Infoprop (Frauenknecht et al., 2025) integrates uncertainty by truncating rollouts using information-theoretic corruption measures, but is not competitive on complex, high-dimensional tasks such as Humanoid-run (see Figure 1). In contrast, we estimate a single predictive uncertainty and weight each synthetic transition accordingly, integrating this directly into the learning objective to preserve useful synthetic data while reducing the influence of uncertain predictions. This yields state-of-the-art results on challenging tasks (Figures 1 and 4).

**Implicit Maximum Likelihood Estimation.** IMLE (Li & Malik, 2018) trains implicit generative models by minimizing the expected distance from each data point to its nearest generated sample, avoiding mode-collapse and GAN (Goodfellow et al., 2014) training issues. Adaptive IMLE (Aghabozorgi et al., 2023) extends this with adaptive thresholding and curriculum learning for better few-shot performance. These methods demonstrate that likelihood-based objectives can achieve good sample quality without adversarial training on low-data settings.

## 6 LIMITATIONS AND FUTURE WORK

WIMLE uses world models solely to generate synthetic rollouts. Other uses, such as planning with the model or integrating the model into policy-gradient formulations, remain unexplored here. Future work should evaluate WIMLE in these settings. Our experiments use proprioceptive state observations only. Extending WIMLE to image-based control is an important direction, especially since IMLE has been shown to be effective in few-shot image synthesis (Aghabozorgi et al., 2023; Vashist et al., 2024). Finally, similar to MBPO (Janner et al., 2019) and POMP (Zhu et al., 2023), the rollout horizon is a task-dependent hyperparameter. Learning to adapt the horizon online based on model confidence is a promising avenue for future research.

## 7 CONCLUSION

WIMLE advances model-based reinforcement learning by extending IMLE to learn stochastic, multi-modal world models and by weighting synthetic data with predictive confidence. This reduces model bias and stabilizes learning while retaining the benefits of synthetic rollouts. Across 40 continuous-control tasks in DMC, MyoSuite, and HumanoidBench, WIMLE achieves superior sample efficiency and competitive or higher asymptotic performance than strong model-free and model-based baselines. Gains are largest on challenging Dog and Humanoid locomotion tasks. On HumanoidBench, WIMLE significantly outperforms baselines and solves 8 of 14 tasks. The approach integrates cleanly with standard RL objectives and scales with compute through ensembles and parallel latent sampling. We hope these results renew interest in practical world models for challenging continuous control.

ACKNOWLEDGMENTS

This research was enabled in part by support provided by NSERC, the BC DRI Group and the Digital Research Alliance of Canada. We are also grateful to James A. Peltier for his support on the computing infrastructure that made this work possible.

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

## AUTHOR STATEMENT: USE OF LANGUAGE MODELS

We used large language models to help polish writing and improve clarity. All ideas, methods, experiments, and analyses were created and verified by the authors. Any suggested text was reviewed and edited by the authors for accuracy and originality.

## A  ALGORITHM DETAILS

Algorithm 3 provides the detailed implementation of WIMLE, including training frequencies, batch sizes, and other practical considerations omitted from the main algorithm for clarity. The IMLE training procedure is detailed in Algorithm 2. Hyperparameters are provided in Section D.

---

**Algorithm 2** IMLE World Model Training

---

1: **Input:** Environment dataset $\mathcal{D}_{\text{env}}$, ensemble $\{g_{\theta_k}\}_{k=1}^K$, number of latent codes $m$, number of updates $U$, learning rate $\eta$
2: **for** $u = 1$ to $U$ **do**
3:     Sample minibatch $\{(s_i, a_i, r_i, s_{i+1})\}_{i \in B}$ with replacement from $\mathcal{D}_{\text{env}}$
4:     Form targets $y_i = [r_i, s_{i+1}]$ for all $i \in B$
5:     **// Assignment Step (Eq. 9)**
6:     Sample $m$ candidate latents $\{z_j\}_{j=1}^m \sim \mathcal{N}(0, I)$
7:     **for** $k = 1$ to $K$ in parallel **do**
8:         $z_{i,k}^\star = \arg\min_{1 \le j \le m} \|g_{\theta_k}(s_i, a_i, z_j) - y_i\|^2$ for all $i \in B$
9:     **// Update Step (Eq. 10)**
10:     **for** $k = 1$ to $K$ in parallel **do**
11:         $\theta_k \leftarrow \theta_k - \eta \nabla_{\theta_k} \frac{1}{|B|} \sum_{i \in B} \|g_{\theta_k}(s_i, a_i, z_{i,k}^\star) - y_i\|^2$

---

**Algorithm 3** WIMLE: Detailed Implementation

---

1: **Input:** Rollout horizon $H$, ensemble size $K = 7$, batch size $B$, model training frequency train_freq, number of latent codes $m$, number of model updates $U$
2: Initialize policy $\pi_\phi$, ensemble of IMLE world models $\{g_{\theta_k}\}_{k=1}^K$, environment dataset $\mathcal{D}_{\text{env}}$, model dataset $\mathcal{D}_{\text{model}}$
3: **for** environment steps **do**
4:     Collect environment transition using $\pi_\phi$; add to $\mathcal{D}_{\text{env}}$
5:     **if** step mod train_freq $= 0$ **then**
6:         **// IMLE World Model Training**
7:         Train ensemble $\{g_{\theta_k}\}_{k=1}^K$ in parallel using Algorithm 2
8:         **// Uncertainty-Aware Rollout Generation**
9:         Clear $\mathcal{D}_{\text{model}}$
10:         Sample batch of starting states $\{s_0^{(i)}\}_{i=1}^B$ with replacement from $\mathcal{D}_{\text{env}}$
11:         **for** $t = 0$ to $H - 1$ **do**
12:           $a_t^{(i)} \sim \pi_\phi(\cdot | s_t^{(i)})$ for all $i \in \{1, \ldots, B\}$
13:           Sample $m$ latents $\{z_j\}_{j=1}^m \sim \mathcal{N}(0, I)$
14:           Generate predictions $\{g_{\theta_k}(s_t^{(i)}, a_t^{(i)}, z_j)\}_{k=1, j=1}^{K, m}$ from all ensemble members for all $i$
15:           Compute predictive uncertainty: $\sigma_t^{(i)} = \text{std}_{k,j}\big[g_{\theta_k}(s_t^{(i)}, a_t^{(i)}, z_j)\big]$ {aggregated over ensembles and latents}
16:           Set weight $w_t^{(i)} = 1/(\sigma_t^{(i)} + 1)$
17:           Select transitions $(s_{t+1}^{(i)}, r_t^{(i)})$ from predictions
18:           Add weighted transitions $\{(s_t^{(i)}, a_t^{(i)}, r_t^{(i)}, s_{t+1}^{(i)}, w_t^{(i)})\}_{i=1}^B$ to $\mathcal{D}_{\text{model}}$
19:     **// Uncertainty-Weighted Policy Learning**
20:     Sample batch from $\mathcal{D}_{\text{env}} \cup \mathcal{D}_{\text{model}}$ (real data has $w = 1$)
21:     Update policy using weighted RL objective:
22:         $\mathcal{L} = \mathbb{E}_{(s,a,r,s',w) \sim \text{batch}}[w \cdot \ell_{\text{RL}}(s, a, r, s')]$

---

## B  THEORETICAL ANALYSIS OF RISK-WEIGHTED BELLMAN ESTIMATION

**Bellman fixed point under positive reweighting.**  Fix a policy $\pi$ and a target value function $V$. Recall the one-step Bellman target

$$y = r + \gamma V(s'), \tag{17}$$

with conditional mean

$$\mu(s,a) = \mathbb{E}[y \mid s,a] = \mathbb{E}_{(r,s') \sim P(\cdot|s,a)}[r + \gamma V(s')]. \tag{18}$$

Given a strictly positive weight function $w : \mathcal{S} \times \mathcal{A} \to (0,\infty)$, the corresponding population weighted Bellman regression loss for an arbitrary action-value function $Q$ is

$$\mathcal{L}_w(Q) = \mathbb{E}_{(s,a) \sim d^\pi}\, \mathbb{E}_{(r,s') \sim P(\cdot|s,a)}\big[w(s,a)\,(y - Q(s,a))^2\big]. \tag{19}$$

**Lemma (Weights do not change the Bellman target).**  Assume that $w(s,a) > 0$ for all $(s,a)$ in the support of $d^\pi$. Then the unique minimizer of $\mathcal{L}_w(Q)$ over all action-value functions $Q : \mathcal{S} \times \mathcal{A} \to \mathbb{R}$ is

$$Q^\star(s,a) = \mu(s,a) \qquad \text{for all } (s,a), \tag{20}$$

which is the same minimizer as for the unweighted objective (obtained by setting $w \equiv 1$).

*Proof.* Fix $(s,a)$ and consider the conditional risk as a function of a scalar $q \in \mathbb{R}$:

$$\ell_{s,a}(q) \;=\; \mathbb{E}\big[\,w(s,a)\,(y-q)^2 \,\big|\, s,a\big] \;=\; w(s,a)\,\mathbb{E}\big[(y-q)^2 \mid s,a\big]. \tag{21}$$

Since $w(s,a)$ is a strictly positive constant with respect to the inner expectation, it does not affect the minimizer in $q$. The derivative of the unweighted term with respect to $q$ is

$$\frac{\partial}{\partial q}\, \mathbb{E}\big[(y-q)^2 \mid s,a\big] \;=\; 2\big(q - \mathbb{E}[y \mid s,a]\big) \;=\; 2\big(q - \mu(s,a)\big), \tag{22}$$

which vanishes if and only if $q = \mu(s,a)$. Thus, for each fixed $(s,a)$, the unique minimizer of $\ell_{s,a}(q)$ is $q = \mu(s,a)$, independently of the choice of $w(s,a) > 0$.

The global objective $\mathcal{L}_w(Q)$ in Eq. 19 is the expectation of $\ell_{s,a}(Q(s,a))$ under $d^\pi$. For any action-value function $Q$, we can write

$$\ell_{s,a}(Q(s,a)) - \ell_{s,a}(\mu(s,a)) = w(s,a)\, \mathbb{E}\big[(Q(s,a) - \mu(s,a))^2 \mid s,a\big] \;\geq\; 0, \tag{23}$$

with equality if and only if $Q(s,a) = \mu(s,a)$. Integrating this inequality with respect to $d^\pi$ shows that $\mathcal{L}_w(Q) \geq \mathcal{L}_w(\mu)$, with strict inequality whenever $Q$ differs from $\mu$ on a set of positive $d^\pi$-measure. Hence, $Q^\star = \mu$ is the unique minimizer of $\mathcal{L}_w$, and this minimizer does not depend on the particular choice of strictly positive weights. $\square$

**Proof of the linear-critic GLS lemma.**  We restate the setting from the main text. Let $x_i = (s_i, a_i)$ denote $m$ state–action pairs with feature vectors $\phi(x_i) \in \mathbb{R}^d$, and let $y_i$ be the corresponding one-step TD targets. Assume a linear value model

$$y_i = \phi(x_i)^\top \theta^\star + \varepsilon_i, \tag{24}$$

where $\theta^\star \in \mathbb{R}^d$ is the true parameter vector, the noise satisfies $\mathbb{E}[\varepsilon_i \mid x_i] = 0$, and the conditional variances are $\mathrm{Var}(\varepsilon_i \mid x_i) = \sigma_i^2$. Collect the targets into a vector $y \in \mathbb{R}^m$, the features into a design matrix $\Phi \in \mathbb{R}^{m \times d}$ whose $i$-th row is $\phi(x_i)^\top$, and define the noise vector $\varepsilon = (\varepsilon_1, \ldots, \varepsilon_m)^\top$. Then

$$y = \Phi\theta^\star + \varepsilon, \tag{25}$$

with $\mathbb{E}[\varepsilon] = 0$ and covariance

$$\Sigma = \mathrm{Cov}(\varepsilon) = \mathrm{diag}(\sigma_1^2, \ldots, \sigma_m^2). \tag{26}$$

We assume that $\Phi$ has full column rank so that all normal equations below are solvable.

**Linear unbiased estimators and GLS.** Consider linear estimators of $\theta^\star$ of the form

$$\hat{\theta} = Ay, \tag{27}$$

for some matrix $A \in \mathbb{R}^{d \times m}$. Unbiasedness for all $\theta^\star$ requires

$$\mathbb{E}[\hat{\theta}] = \mathbb{E}[A(\Phi\theta^\star + \varepsilon)] = A\Phi\theta^\star = \theta^\star \quad \text{for all } \theta^\star, \tag{28}$$

which is equivalent to the constraint

$$A\Phi = I_d. \tag{29}$$

Under this constraint, the covariance of $\hat{\theta}$ is

$$\mathrm{Cov}(\hat{\theta}) = A \, \mathrm{Cov}(y) \, A^\top = A\Sigma A^\top. \tag{30}$$

The generalized least-squares (GLS) estimator corresponds to the specific choice

$$A_{\mathrm{GLS}} = \left(\Phi^\top \Sigma^{-1} \Phi\right)^{-1} \Phi^\top \Sigma^{-1}, \tag{31}$$

so that

$$\hat{\theta}_{\mathrm{GLS}} = A_{\mathrm{GLS}} y. \tag{32}$$

It is immediate that $A_{\mathrm{GLS}}\Phi = I_d$, so $\hat{\theta}_{\mathrm{GLS}}$ is linear and unbiased. Its covariance is

$$\mathrm{Cov}(\hat{\theta}_{\mathrm{GLS}}) = A_{\mathrm{GLS}}\Sigma A_{\mathrm{GLS}}^\top = \left(\Phi^\top \Sigma^{-1} \Phi\right)^{-1}. \tag{33}$$

**Optimality of GLS.** Let $\tilde{\theta} = Ay$ be any other linear unbiased estimator, so that $A\Phi = I_d$ by equation 29. Define

$$C = A - A_{\mathrm{GLS}}. \tag{34}$$

Then

$$C\Phi = A\Phi - A_{\mathrm{GLS}}\Phi = I_d - I_d = 0. \tag{35}$$

The covariance of $\tilde{\theta}$ can be expanded as

$$\mathrm{Cov}(\tilde{\theta}) = A\Sigma A^\top \tag{36}$$

$$= (A_{\mathrm{GLS}} + C) \Sigma (A_{\mathrm{GLS}} + C)^\top \tag{37}$$

$$= A_{\mathrm{GLS}}\Sigma A_{\mathrm{GLS}}^\top + A_{\mathrm{GLS}}\Sigma C^\top + C\Sigma A_{\mathrm{GLS}}^\top + C\Sigma C^\top. \tag{38}$$

We now show that the cross terms vanish. Using the definition of $A_{\mathrm{GLS}}$,

$$A_{\mathrm{GLS}}\Sigma = \left(\Phi^\top \Sigma^{-1}\Phi\right)^{-1}\Phi^\top\Sigma^{-1}\Sigma = \left(\Phi^\top\Sigma^{-1}\Phi\right)^{-1}\Phi^\top, \tag{39}$$

and similarly

$$\Sigma A_{\mathrm{GLS}}^\top = \Sigma\Sigma^{-1}\Phi\left(\Phi^\top\Sigma^{-1}\Phi\right)^{-1} = \Phi\left(\Phi^\top\Sigma^{-1}\Phi\right)^{-1}. \tag{40}$$

Therefore,

$$A_{\mathrm{GLS}}\Sigma C^\top = \left(\Phi^\top\Sigma^{-1}\Phi\right)^{-1}\Phi^\top C^\top = \left(\Phi^\top\Sigma^{-1}\Phi\right)^{-1}(C\Phi)^\top = 0, \tag{41}$$

$$C\Sigma A_{\mathrm{GLS}}^\top = C\Phi\left(\Phi^\top\Sigma^{-1}\Phi\right)^{-1} = 0, \tag{42}$$

where we used equation 35. Thus

$$\mathrm{Cov}(\tilde{\theta}) = \mathrm{Cov}(\hat{\theta}_{\mathrm{GLS}}) + C\Sigma C^\top. \tag{43}$$

For any vector $v \in \mathbb{R}^d$,

$$v^\top C\Sigma C^\top v = (C^\top v)^\top \Sigma (C^\top v) \geq 0, \tag{44}$$

because $\Sigma$ is positive semidefinite. Hence $C\Sigma C^\top$ is positive semidefinite, and we have

$$\mathrm{Cov}(\hat{\theta}_{\mathrm{GLS}}) \preceq \mathrm{Cov}(\tilde{\theta}), \tag{45}$$

with equality if and only if $C = 0$, i.e., $A = A_{\mathrm{GLS}}$. This establishes that GLS has minimum covariance among all linear unbiased estimators.

**Connection to inverse-variance weighting.** In our setting, weighted least squares with per-sample weights $w_i > 0$ corresponds to minimizing

$$\sum_{i=1}^{m} w_i \big(y_i - \phi(x_i)^\top \theta\big)^2 = (y - \Phi\theta)^\top W (y - \Phi\theta), \tag{46}$$

where $W = \mathrm{diag}(w_1, \ldots, w_m)$. The normal equations yield the estimator

$$\hat{\theta}_W = (\Phi^\top W \Phi)^{-1} \Phi^\top W y. \tag{47}$$

Choosing weights $w_i \propto 1/\sigma_i^2$ makes $W$ proportional to $\Sigma^{-1}$, so that

$$\hat{\theta}_W = (\Phi^\top \Sigma^{-1} \Phi)^{-1} \Phi^\top \Sigma^{-1} y = \hat{\theta}_{\mathrm{GLS}}. \tag{48}$$

Thus inverse-variance weights $w_i \propto 1/\sigma_i^2$ recover the GLS estimator and, by the argument above, minimize the covariance matrix among all linear unbiased estimators. $\square$

## C  PER-TASK RESULTS

We present detailed per-task performance results for WIMLE and other baselines across all benchmarks. The performance on each individual task is shown in Figures 7, 8, 9, and 10.

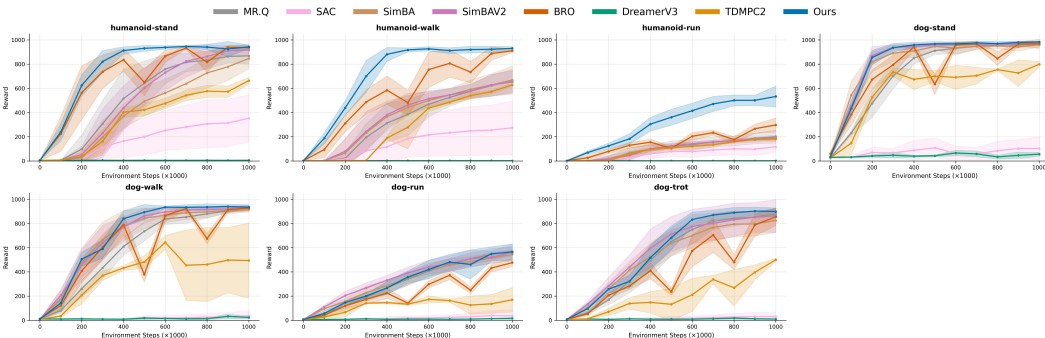

Figure 7: Per-task results for high-dimensional Dog & Humanoid tasks from DeepMind Control Suite. We present the IQM of rewards and $95\%$ confidence intervals for BRO and other baselines run for $1M$ steps.

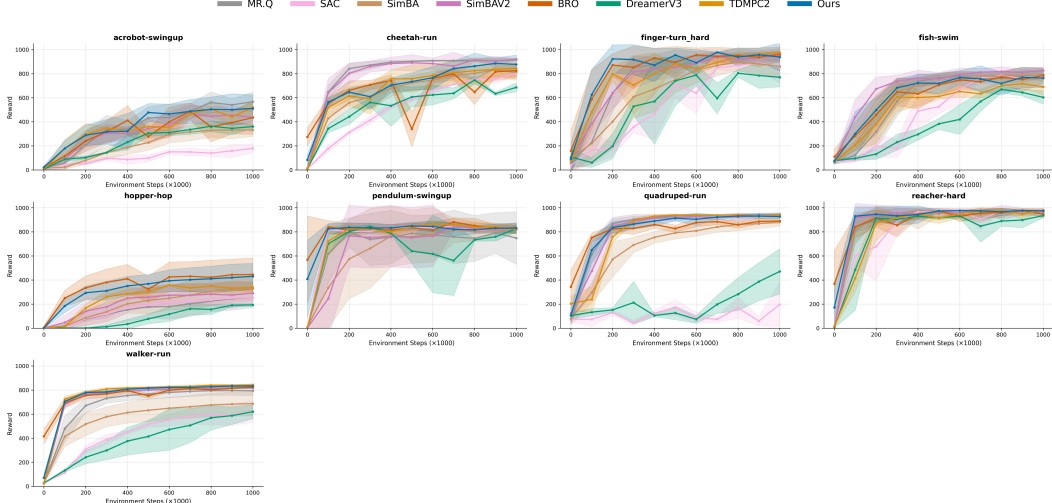

Figure 8: Per-task results for DeepMind Control Suite tasks with low-dimensional state/action spaces. We present the IQM of rewards and $95\%$ confidence intervals for WIMLE and other baselines run for $1M$ steps.

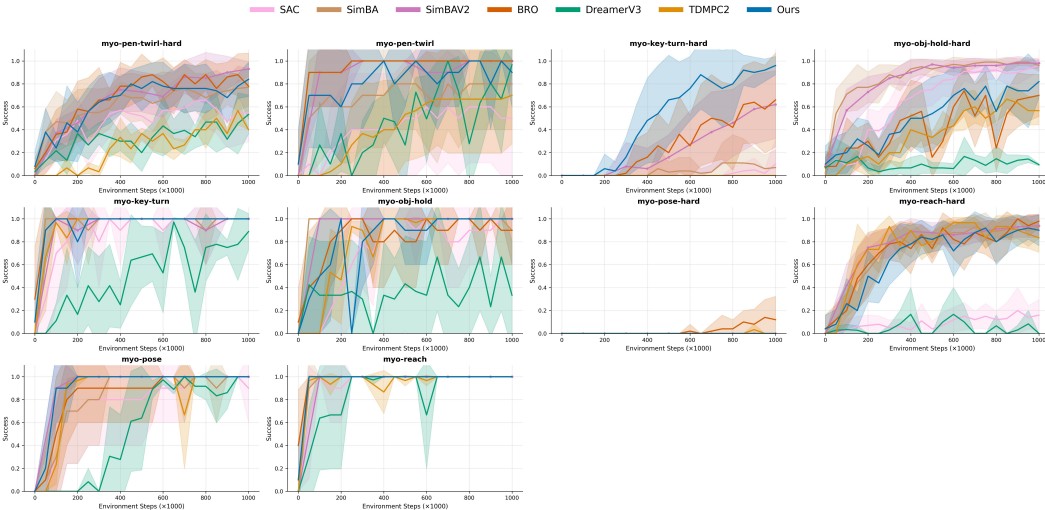

Figure 9: Per-task results for MyoSuite tasks. We present the IQM of success rate and $95\%$ confidence intervals for WIMLE and other baselines run for $1M$ steps.

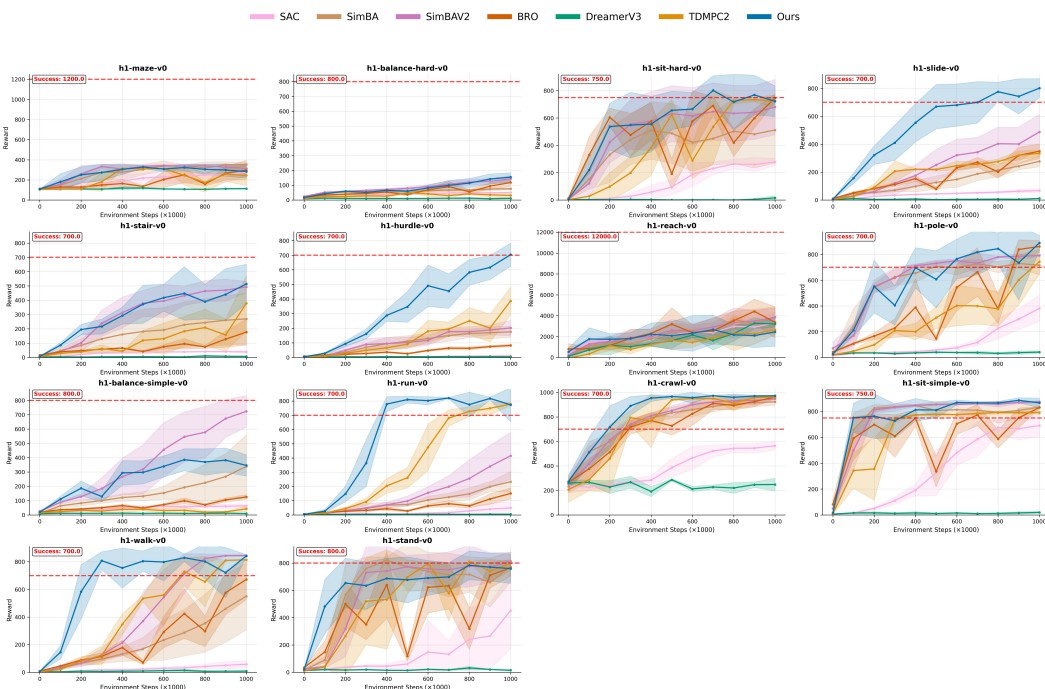

Figure 10: Per-task results for HumanoidBench tasks. We present the IQM of rewards and $95\%$ confidence intervals for WIMLE and other baselines run for $1M$ steps. The red dashed line indicates the success threshold for each task.

## D HYPERPARAMETERS

Table 1 lists the common hyperparameters used across all tasks. These parameters were selected through hyperparameter tuning based on standard practices in RL.

Based on our empirical evaluations, we found that increasing the model batch size to the maximum extent allowed by available GPU resources while proportionally decreasing the number of model

Table 1: Common hyperparameters used across all tasks.

| Parameter | Value |
|---|---|
| **SAC Parameters** | |
| Batch size | 128 |
| Actor learning rate | $3 \times 10^{-4}$ |
| Critic learning rate | $3 \times 10^{-4}$ |
| Number of quantiles | 100 |
| Updates per step | 10 |
| **World Model Parameters** | |
| Model learning rate | $1 \times 10^{-3}$ |
| Model batch size | 512 |
| Model updates | 200 |
| Number of latent codes | 4 |
| Model training frequency | 1000 |
| Number of rollouts | 300 |
| Number of ensembles | 7 |

updates can achieve similar performance with improved training speed. When scaling the batch size in this manner, the model learning rate should be adjusted accordingly following standard Machine Learning practices.

### D.1 ROLLOUT LENGTH SELECTION

We select task-specific rollout horizons H through experimentation. For easier tasks where base-lines already saturate near the maximum score (e.g., MyoSuite manipulation tasks), we start with short horizons (H=1-2) and increase only if performance benefits are observed, as longer horizons may still introduce slight performance degradation—though not to the extent seen in traditional MBRL methods. For harder tasks requiring longer-term planning (e.g., HumanoidBench, Dog & Humanoid), we begin with longer horizons (H=8) and decrease only if performance gains are seen empirically. However, we note that even simpler tasks may benefit from longer horizons in some cases, reflecting the task-specific nature of optimal rollout length. This selection balances the benefits of synthetic data augmentation with computational cost, as the marginal benefit of additional rollout steps diminishes beyond each task's optimal horizon. We cap $H$ at 8 to maintain rollout throughput and because we observe diminishing returns beyond task-specific optima; Tables 6, 7, and 8 report the chosen $H$ per task. An interesting future direction would be to dynamically adjust rollout horizons based on the model's uncertainty level, potentially allowing for adaptive rollout lengths that scale with model confidence.

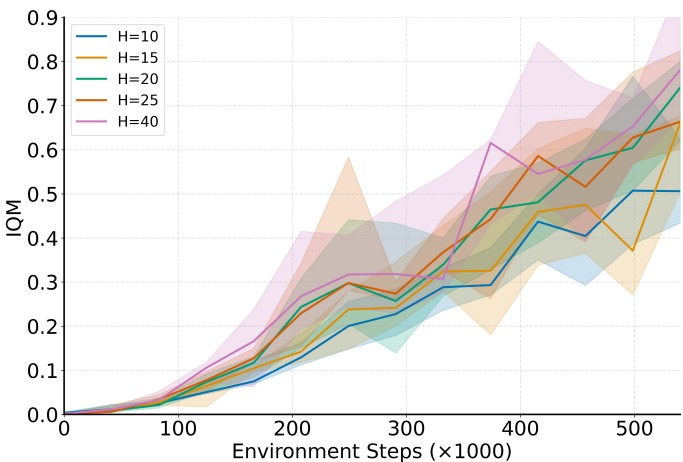

Figure 11: Rollout length sensitivity on h1-hurdle-v0 task of HumanoidBench up to $H = 40$. WIMLE remains stable at longer horizons without rollout scheduling.

## E    EXPERIMENT DETAILS

This section provides detailed descriptions of the benchmark environments used in our evaluation. We explain the task suites, their characteristics, and the normalization procedures used for fair comparison across different score scales. The following subsections describe each benchmark suite with complete task lists and their state/action dimensions.

### E.1    DETAILED RESULTS

We present comprehensive IQM results for WIMLE and baseline methods across all benchmark suites at 100k, 200k, 500k, and 1M environment steps. The best performing method for each step count is highlighted in bold and the second best are underlined. **WIMLE** performs better across most evaluations.

Table 2: IQM results for DMC suite. Best scores are highlighted in bold, second best are underlined.

| Method | 100k | 200k | 500k | 1M |
|---|---|---|---|---|
| MR.Q | 0.153 | 0.362 | 0.714 | 0.830 |
| SAC | 0.037 | 0.082 | 0.210 | 0.326 |
| SimBA | 0.120 | 0.263 | 0.522 | 0.691 |
| SimBAV2 | 0.235 | 0.495 | 0.730 | 0.845 |
| BRO | 0.294 | 0.519 | 0.542 | 0.846 |
| DreamerV3 | 0.051 | 0.075 | 0.165 | 0.286 |
| TD-MPC2 | 0.152 | 0.374 | 0.566 | 0.696 |
| **WIMLE** | **0.332** | **0.575** | **0.812** | **0.871** |

### E.2    DEEPMIND CONTROL SUITE

DeepMind Control Suite (Tassa et al., 2018, DMC) is a standard continuous control benchmark encompassing locomotion and manipulation tasks with varying complexity. We evaluate 16 tasks from DMC, focusing on the most challenging locomotion tasks including Dog and Humanoid embodiments. All returns are normalized by dividing by 1000 to scale performance to [0,1]. The complete list of tasks with their observation and action dimensions is provided in Table 6.

Table 3: IQM results for Dog & Humanoid suite. Best scores are highlighted in bold, second best are underlined.

| Method | 100k | 200k | 500k | 1M |
|---|---|---|---|---|
| MR.Q | 0.042 | 0.127 | 0.557 | 0.796 |
| SAC | 0.007 | 0.008 | 0.043 | 0.069 |
| SimBA | 0.067 | 0.173 | 0.533 | 0.773 |
| SimBAV2 | 0.082 | 0.200 | 0.601 | 0.808 |
| BRO | 0.086 | 0.290 | 0.355 | 0.864 |
| DreamerV3 | 0.006 | 0.006 | 0.007 | 0.010 |
| TD-MPC2 | 0.014 | 0.058 | 0.302 | 0.527 |
| **WIMLE** | **0.140** | **0.389** | **0.803** | **0.897** |

Table 4: IQM results for MyoSuite. Best scores are highlighted in bold, second best are underlined.

| Method | 100k | 200k | 500k | 1M |
|---|---|---|---|---|
| SAC | 0.038 | 0.350 | 0.622 | 0.714 |
| SimBA | 0.566 | 0.728 | 0.912 | 0.952 |
| SimBAV2 | **0.724** | **0.830** | **0.956** | **0.990** |
| BRO | 0.440 | 0.736 | 0.816 | 0.980 |
| DreamerV3 | 0.028 | 0.044 | 0.181 | 0.466 |
| TD-MPC2 | 0.088 | 0.394 | 0.688 | 0.775 |
| **WIMLE** | 0.460 | 0.620 | 0.928 | 0.980 |

Table 5: IQM results for HumanoidBench. Best scores are highlighted in bold, second best are underlined.

| Method | 100k | 200k | 500k | 1M |
|---|---|---|---|---|
| SAC | 0.008 | 0.020 | 0.060 | 0.168 |
| SimBA | 0.070 | 0.164 | 0.322 | 0.521 |
| SimBAV2 | 0.059 | 0.179 | 0.488 | 0.799 |
| BRO | 0.064 | 0.127 | 0.100 | 0.530 |
| DreamerV3 | 0.003 | 0.003 | 0.005 | 0.007 |
| TD-MPC2 | 0.023 | 0.064 | 0.382 | 0.734 |
| **WIMLE** | **0.169** | **0.393** | **0.715** | **0.898** |

### E.3 MYOSUITE

MyoSuite (Caggiano et al., 2022) provides high-fidelity musculoskeletal simulations for dexterous manipulation tasks. We evaluate 10 tasks including both fixed-goal and randomized-goal (hard) settings. Performance is measured using success rates, which naturally scale to [0,1]. The complete list of tasks with their observation and action dimensions is provided in Table 7.

### E.4 HUMANOIDBENCH

HumanoidBench (Sferrazza et al., 2024) provides locomotion tasks for the UniTree H1 humanoid robot. We evaluate 14 tasks spanning balance, locomotion, and manipulation. For fair comparison across tasks with different score scales, all HumanoidBench scores are normalized using each task's target success score and random score following the same procedure as in (Lee et al., 2025a;b):

$$\text{Success-Normalized}(x) := \frac{x - \text{random score}}{\text{Target success score} - \text{random score}}$$

The complete list of tasks with their observation and action dimensions is provided in Table 8, and the random scores and target success scores used for normalization are listed in Table 9.

Table 6: **DMC Tasks.** Complete list of 16 DMC tasks evaluated, with state and action dimensions and rollout lengths.

| Task | State dim $|\mathcal{S}|$ | Action dim $|\mathcal{A}|$ | H |
|---|---|---|---|
| acrobot-swingup | 6 | 1 | 8 |
| cheetah-run | 17 | 6 | 1 |
| finger-turn_hard | 12 | 2 | 1 |
| fish-swim | 24 | 5 | 8 |
| hopper-hop | 15 | 4 | 1 |
| pendulum-swingup | 3 | 1 | 1 |
| quadruped-run | 78 | 12 | 2 |
| reacher-hard | 6 | 2 | 1 |
| walker-run | 24 | 6 | 1 |
| humanoid-stand | 67 | 24 | 2 |
| humanoid-walk | 67 | 24 | 6 |
| humanoid-run | 67 | 24 | 6 |
| dog-stand | 223 | 38 | 6 |
| dog-walk | 223 | 38 | 4 |
| dog-run | 223 | 38 | 6 |
| dog-trot | 223 | 38 | 4 |

Table 7: **MyoSuite Tasks.** Complete list of 10 MyoSuite tasks evaluated, with state and action dimensions and rollout lengths.

| Task | State dim $|\mathcal{S}|$ | Action dim $|\mathcal{A}|$ | H |
|---|---|---|---|
| myo-key-turn | 93 | 39 | 6 |
| myo-key-turn-hard | 93 | 39 | 1 |
| myo-obj-hold | 91 | 39 | 4 |
| myo-obj-hold-hard | 91 | 39 | 1 |
| myo-pen-twirl | 83 | 39 | 2 |
| myo-pen-twirl-hard | 83 | 39 | 1 |
| myo-pose | 108 | 39 | 2 |
| myo-pose-hard | 108 | 39 | 1 |
| myo-reach | 115 | 39 | 4 |
| myo-reach-hard | 115 | 39 | 1 |

Table 8: **HumanoidBench Tasks.** Complete list of 14 HumanoidBench tasks evaluated, with state and action dimensions and rollout lengths.

| Task | State dim $|\mathcal{S}|$ | Action dim $|\mathcal{A}|$ | H |
|---|---|---|---|
| h1-balance_simple-v0 | 64 | 19 | 6 |
| h1-balance_hard-v0 | 77 | 19 | 6 |
| h1-crawl-v0 | 51 | 19 | 10 |
| h1-hurdle-v0 | 51 | 19 | 10 |
| h1-maze-v0 | 51 | 19 | 10 |
| h1-pole-v0 | 51 | 19 | 10 |
| h1-reach-v0 | 57 | 19 | 10 |
| h1-run-v0 | 51 | 19 | 10 |
| h1-slide-v0 | 51 | 19 | 10 |
| h1-sit_hard-v0 | 51 | 19 | 6 |
| h1-sit_simple-v0 | 51 | 19 | 10 |
| h1-stair-v0 | 51 | 19 | 10 |
| h1-stand-v0 | 51 | 19 | 10 |
| h1-walk-v0 | 51 | 19 | 10 |

Table 9: **HumanoidBench Normalization Scores.** Random scores and target success scores used for normalization.

| Task | Random Score | Target Success Score |
|---|---|---|
| h1-balance-simple | 9.391 | 800 |
| h1-balance-hard | 9.044 | 800 |
| h1-crawl | 272.658 | 700 |
| h1-hurdle | 2.214 | 700 |
| h1-maze | 106.441 | 1200 |
| h1-pole | 20.09 | 700 |
| h1-reach | 260.302 | 12000 |
| h1-run | 2.02 | 700 |
| h1-slide-v0 | 2.02 | 700 |
| h1-slide-v1 | 2.02 | 700 |
| h1-sit-hard | 10.545 | 800 |
| h1-stair | 2.214 | 700 |
| h1-stand | 10.545 | 800 |
| h1-walk | 2.377 | 700 |

# F  ADDITIONAL RESULTS

Figure 12 compares standard WIMLE (mixed real + imagined data) against an "imagined-only" variant whose critic and actor are trained exclusively on model-generated rollouts. Removing real transitions does reduce performance slightly, but the imagined-only curve remains close to the original WIMLE results, underscoring that our synthetic trajectories are strong enough to sustain competitive learning.

Figure 14 and Table 10 illustrate the wall-clock efficiency of WIMLE. Despite the overhead of ensemble training, WIMLE's superior sample efficiency results in a significantly lower total time-to-solution compared to baselines.

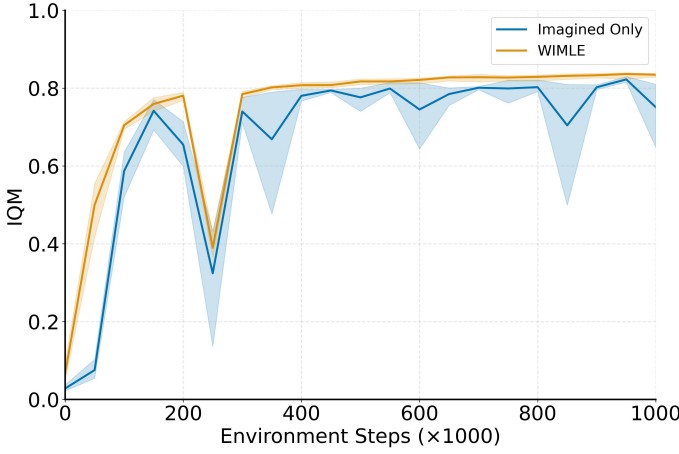

Figure 12: Imagined-only WIMLE (no real data in critic/actor updates) achieves performance comparable to the standard configuration, highlighting the strength of the generated rollouts.

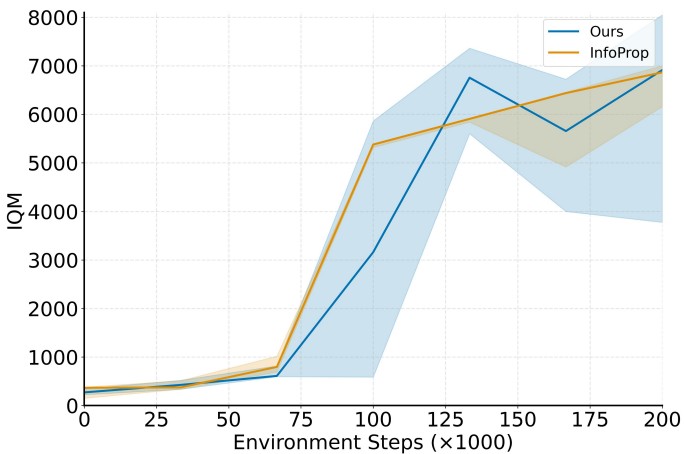

Figure 13: Humanoid (Mujoco) comparison against InfoProp for 4 seeds. Unlike InfoProp, which uses task-specific termination functions for synthetic rollouts, WIMLE runs with the generic setup used for our DMC experiments yet still matches or exceeds InfoProp.

| Method | Step 300k | | Step 500k | | Step 1000k | |
|---|---|---|---|---|---|---|
| | Time (h) | IQM | Time (h) | IQM | Time (h) | IQM |
| Infoprop | 11.49 | 0.044 | 19.15 | 0.102 | 38.30 | 0.077 |
| DreamerV3 | 3.90 | 0.001 | 6.50 | 0.001 | 13.00 | 0.001 |
| TD-MPC2 | 3.43 | 0.001 | 5.72 | 0.038 | 11.43 | 0.190 |
| Ours | 4.74 | 0.168 | 7.91 | 0.345 | 15.81 | 0.561 |

Table 10: Performance and time at different training milestones.

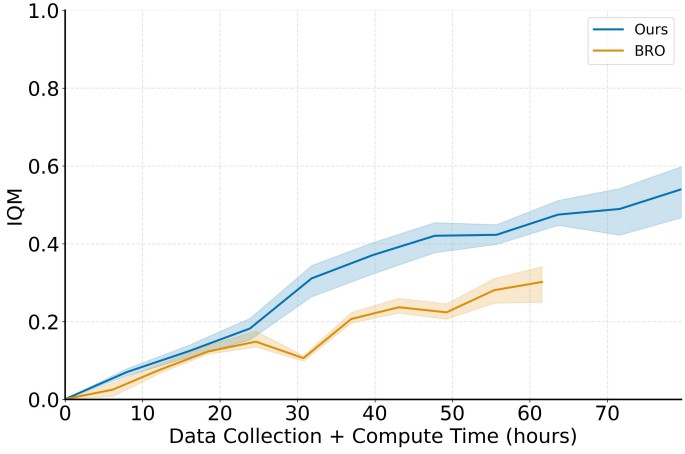

Figure 14: Projected total wall-clock time (algorithm compute + data collection) on Humanoid-run, assuming a 5Hz control rate. WIMLE reaches asymptotic performance significantly faster than BRO (the fastest model-free baseline) because its superior sample efficiency drastically reduces the time spent collecting real-world data, outweighing its higher per-update compute cost.

