# OpenReview forum: "WIMLE: Uncertainty‑Aware World Models with IMLE for Sample‑Efficient Continuous Control"
_ICLR.cc/2026/Conference — ICLR 2026 Poster_

### Official Review · Reviewer_z1Wb · 2025-10-19

**Soundness:** 3
**Presentation:** 3
**Contribution:** 3
**Rating:** 4
**Confidence:** 3

**Summary:**

This paper proposes WIMLE, a Model-Based Reinforcement Learning (MBRL) approach that constructs an uncertainty-aware world model by integrating the IMLE generative model. The method aims to address compound errors and the issue of overconfident predictions prevalent in existing MBRL algorithms. The authors evaluate WIMLE on several vector-state benchmarks, including DMC, HumanoidBench, and MyoSuite. The results indicate that WIMLE achieves significant improvements in both sample efficiency and asymptotic performance.

**Strengths:**

- The paper successfully extends and integrates IMLE (a mode-covering generative model proven effective in low-data regimes) into the MBRL framework for world model learning.
- The method effectively integrates uncertainty estimates from model predictions into the Reinforcement Learning objective function, which helps mitigate policy misalignment caused by model bias and overconfident predictions.
- WIMLE demonstrates substantial improvements over strong model-free and model-based baselines across multiple continuous control tasks, particularly showcasing outstanding performance on the challenging Humanoid-run task.

**Weaknesses:**

- Limitation in Observation Space: The experimental evaluation is limited to vector-state (low-dimensional) environments. Many state-of-the-art MBRL algorithms have demonstrated strong performance in high-dimensional (pixel/image) state spaces. The paper currently lacks verification of WIMLE’s performance in these crucial high-dimensional settings.
- Potential for Unfair Comparison: The WIMLE algorithm uses a mixture of real and imaginary data for policy training. In contrast, the original paper implementations of comparable baselines (e.g., DreamerV3, TDMPC2) typically rely solely on imaginary trajectories. This difference in data usage introduces a potential source of unfairness in the comparison and requires clarification from the authors.

**Questions:**

1.  As noted in Weakness 1, could the experimental scope be extended to image-state environments (e.g., a subset of DMC tasks using raw pixel observations or the Atari 100k benchmark) to evaluate WIMLE’s effectiveness in such contexts?
2.  As noted in Weakness 2, given that the comparison algorithms (DreamerV3, TDMPC2) primarily use model-generated trajectories for policy training in their original implementations, please clarify whether this variable was unified in the current comparative experiments. If not, to ensure a fair comparison, could the authors provide an ablation experiment where the WIMLE policy is trained exclusively on model-generated trajectories (i.e., without real environment data)?
3. WIMLE uses uncertainty-based weighted updates within the Actor-Critic architecture. Is the weighting factor applied only to the loss update of the Value Network, or is it applied to the loss updates of both the Policy Network and the Value Network?
4. It is recommended that the results presented in Figure 2 be converted into a table format. The current graphical representation is poor for clearly depicting the performance (e.g., time efficiency) of lower-performing algorithms like DreamerV3.
5. Code and Implementation Details: To ensure research reproducibility, can the authors publicly release the complete experimental code and model implementation upon acceptance of the paper? This is critical for the community to verify the claimed performance improvements.

---

> ### Author Response · Authors · 2025-11-21
>
> ## Q1: Mixture of real and imagined data versus pure imagination training
> We thank the reviewer for raising this point. To address it, we trained an 'imagined-only' WIMLE variant where the actor and critic are trained exclusively on model-generated trajectories. As shown in Figure 12 (Appendix), this variant achieves performance comparable to the standard WIMLE, confirming the robustness of our results. Regarding baselines, DreamerV3 trains solely on imagined data to avoid off-policy corrections due to its REINFORCE-style updates, whereas TD-MPC2 uses its model for planning. WIMLE follows the MBPO recipe, which is designed to mix real and synthetic data effectively.
>
> ## Q2: Code release and reproducibility
> We appreciate the emphasis on reproducibility. We will release the full codebase, configuration files, and pretrained checkpoints upon acceptance.
>
> ## Q3: Where are uncertainty weights applied?
> The weighting is applied only to the critic's TD objective. We also explored applying it to the actor objective in preliminary experiments, but observed no additional gains. Thus, we apply uncertainty weights solely to the critic to maintain simplicity.
>
> ## Q4: Figure 2 readability
> We thank the reviewer for this suggestion. We have converted the results to a tabular format as requested and included them in Table 10 (Appendix) to clearly present the performance and efficiency metrics for all methods.
>
> ## Q5: Scope of state-based experiments and extension to visual settings
> We thank the reviewer for this suggestion. We emphasize that our primary contribution (uncertainty-aware world models) is orthogonal to input modality. We chose our current benchmarks (DMC, MyoSuite, HumanoidBench) as they are standard in the literature for evaluating continuous control complexity and contact-rich dynamics with proprioceptive state information [1][2][3][4][5], which is critical for real-world robotics where robust state estimates are often available. We acknowledge the importance of visual control. We are currently adapting our architecture for pixel-based tasks (e.g., Atari) and will include a discussion in the camera-ready version.
>
> [1] Colas et al., Bigger, Regularized, Optimistic: scaling for compute and sample-efficient continuous control, 2024.
>
> [2] Lee et al., Hyperspherical Normalization for Scalable Deep Reinforcement Learning, 2025.
>
> [3] Lee et al., SimBa: Simplicity Bias for Scaling Up Parameters in Deep Reinforcement Learning, ICLR 2025.
>
> [4] Frauenknecht et al., On rollouts in model-based reinforcement learning. In Proceedings of the 2025 International Conference on Learning Representations (ICLR 2025). arXiv preprint arXiv:2501.16918. Available at https://arxiv.org/abs/2501.16918
>
> [5] Frauenknecht et al., Trust the model where it trusts itself: Model-based actor-critic with uncertainty-aware rollout adaption. In Proceedings of the 41st International Conference on Machine Learning (ICML 2024) (Vol. 235, pp. 13973–14005). Available at https://arxiv.org/abs/2405.19014

---

> ### Comment · Reviewer_z1Wb · 2025-11-26
>
> Thank you to the authors for the detailed response to my questions.
>
> I am very pleased to see that the authors have included the "imagined-only" WIMLE variant in the Appendix and demonstrated that its performance is comparable to the standard WIMLE. This effectively addresses my concern regarding the "potential for unfair comparison."
>
> However, while the authors have promised in their reply to include a relevant discussion (or preliminary results) on the validation in high-dimensional (pixel/image) environments in the final version, I will temporarily maintain the current rating until I see this discussion or preliminary experimental results. I look forward to reading the discussion on WIMLE's potential applicability and the challenges it faces in visual control tasks in the camera-ready version.

---

> > ### Author Response · Authors · 2025-12-03
> >
> > We thank the reviewer for their positive feedback on our "imagined-only" experiment.
> >
> > To address your remaining point on visual control and demonstrate the generality of our approach, we have integrated WIMLE into SPR, a standard data-efficient method for Atari 100k [1]. In preliminary experiments on the games *Krull* and *Qbert* (4 seeds), we find that WIMLE+SPR is both more sample-efficient and achieves higher final performance than SPR on these pixel-based tasks.
> >
> > As summarized below, on Krull WIMLE+SPR reaches SPR's final 100k score of ~3688.9 at 30k environment steps (about **3×** more sample-efficient) and attains a higher final score of 4650.3 at 100k steps. On Qbert, WIMLE+SPR reaches SPR's final 100k score of ~669.1 at 60k steps (about **1.7×** more sample-efficient) and improves the final score to 776.0. These preliminary Atari 100k results suggest that WIMLE's benefits can transfer to pixel-based environments. The table below reports the corresponding scores and sample-efficiency gains.
> >
> > | Game  | Method    | Final score @ 100k | Steps to reach SPR 100k score | Relative sample efficiency |
> > | :---- | :-------- | :----------------: | :----------------------------: | :------------------------: |
> > | Krull | SPR       | 3688.9             | 100k                           | 1.0×                       |
> > | Krull | WIMLE+SPR | 4650.3             | 30k                            | ≈3.0×                      |
> > | Qbert | SPR       | 669.1              | 100k                           | 1.0×                       |
> > | Qbert | WIMLE+SPR | 776.0              | 60k                            | ≈1.7×                      |
> >
> > [1] Schwarzer et al., Data-Efficient Reinforcement Learning with Self-Predictive Representations, ICLR 2021.

---

### Official Review · Reviewer_8Fgq · 2025-10-24

**Soundness:** 2
**Presentation:** 2
**Contribution:** 3
**Rating:** 6
**Confidence:** 4

**Summary:**

The paper proposed to leverage Implicit Maximum Likelihood Estimation (IMLE) as base architecture for model-based reinforcement learning.

While the use of IMLE is to some extent incremental, leveraging its uncertainty estimates within TD learning is interesting.
Overall the empirical results of this paper are encouraging: on most tasks the performance generally modest, however on several harder tasks (Humanoid Run, H1-slide and Myo-key-turn-hard), the proposed algorithm is roughly two to three times more sample efficient.

**Strengths:**

* Weighting Bellman backups by model uncertainty is an interesting idea.

* The experiments are convincing: the paper compares WIMLE across several hard continuous RL benchmarks against what is (to my knowledge) the state-of-the-art algorithms (model-free and model-based)

**Weaknesses:**

**The usage of uncertainty is not well grounded**
* For instance, equation 12 does not distinguish between the aleatoric and epistemic uncertainty. This can lead to low credit assignment in states with high stochasticity, rather than states with large model error. I believe this is not observed in the experiments as most of these Mujoco tasks are mostly deterministic.
* In addition, rewarding states based on their epistemic uncertainty (“intrinsic rewards”) is known to accelerate learning in sparse reward tasks. A discussion/study on how WIMLE works even in light of this contradiction would improve the paper.
* Aside from very high-level intuition about penalizing states where model accuracy is low, there is no grounding for why this may work. Building some theoretical intuition (even if not at the strictest rigorousness) would improve the paper.

**Other**

* Minor: adding vision tasks would make the contribution of this paper more impactful.

**Questions:**

* What UTD values did you use when comparing the model-free methods (e.g. BRO)? Did you try to ablate their performance and sample efficiency when increasing UTD?
* How does WIMLE compares to model-free methods in terms of wall-clock time? It would be great to discuss/demonstrate empirically the tradeoff between performance and wall-clock time.

---

> ### Author Response · Authors · 2025-11-21
>
> ## Q1: Predictive uncertainty (aleatoric vs. epistemic) and relation to exploration bonuses
> We thank the reviewer for this insightful question. We have now explicitly decomposed the predictive uncertainty in Section 3.2 by the law of total variance. Eq. 13 in the manuscript shows that our estimator captures both epistemic disagreement and aleatoric/multi-modal variation. Following the decision-theoretic view in [1], the *total* predictive variance represents the Bayes risk of acting on a synthetic transition under a squared-error loss. Therefore, we use this total uncertainty to weight the TD objective for synthetic data, which mitigates bias from unreliable predictions. We emphasize that exploration is orthogonal to our method. Exploration bonuses can be used to incentivize the policy to visit uncertain states. Our method downweights unreliable synthetic data to prevent it from hindering learning.
>
> ## Q2: Theoretical grounding for uncertainty weighting
> We agree that theoretical grounding is valuable. In addition to the Bayes risk perspective discussed above and in [1], WIMLE performs a risk-weighted regression: by mapping the estimated risk $\\sigma^2$ to weights $w \\in (0,1]$, we down-weight high-risk synthetic transitions. This aligns with classical heteroscedastic regression principles, where data points are weighted inversely to their variance. We thank the reviewer for encouraging us to provide this intuition and are happy to clarify further if needed.
>
> ## Q3: Empirical tradeoff between performance and wall-clock time
> We thank the reviewer for this suggestion. We have included a direct wall-clock comparison in Figure 14 (Appendix) against BRO, the fastest model-free baseline. Following [2], we account for total time by including real-world data collection costs (e.g., robot operation, human supervision). Our results show that WIMLE is significantly more efficient than model-free baselines in total time-to-solution as it achieves high performance with far fewer environment samples, outweighing the additional computational cost of its world model.
>
> ## Q4: Update-to-data ratios for model-free baselines
> We strictly followed the recommended hyperparameters for all baselines. WIMLE uses UTD=10 following BRO's finding that gains are limited beyond this value. We kept UTD fixed to the authors' best settings for all methods.
>
> [1] Smith et al. Rethinking Aleatoric and Epistemic Uncertainty. CoRR, abs/2412.20892, 2024.
>
> [2] Zhou et al., Real World Offline Reinforcement Learning with Realistic Data Source, ICRA 2023.

---

> ### Comment · Reviewer_8Fgq · 2025-11-22
>
> Thank you for your detailed answer!
>
> I am still not fully convinced by your clarification regarding the predictive uncertainty. Let me describe your claim based on my understanding.
>
> > our estimator captures both epistemic disagreement and aleatoric/multi-modal variation
>
> Exactly, your estimator captures both of them, however you do not decompose them when weighting the TD-error loss.
>
> > Following the decision-theoretic view in [1], the total predictive variance represents the Bayes risk of acting on a synthetic transition under a squared-error loss. Therefore, we use this total uncertainty to weight the TD objective for synthetic data, which mitigates bias from unreliable predictions.
>
> That is a valid choice. Using the _total_ predictive variance allows you to penalize TD-errors based on how “noisy” the next state prediction is. As mentioned above (and in the paper cited), this noise originates from two sources: **(1)** “unreliable” model predictions, and **(2)** the fact that the dynamics are inherently stochastic, which is (hopefully) captured by IMLE. Suppose that instead of using your learned model, you had access to a perfect model of the dynamics from which you could accurately sample next-states at will. What will your TD-error weighting scheme do? It will reduce the weight of those states that are _inherently_ noisy because of the MDP dynamics, _irrespective of how good the model_ is. In the limit, this implies that in highly stochastic dynamics, your value function will have very weak learning signal in a large portion of the states.
> I believe the reason that this is not problematic in practice is because most of the Mujoco benchmarks are deterministic,
> therefore prediction errors _dominate_ dynamics stochasticity. On the other hand, this seems like a very easy fix; Equations 11 and 12 already show that you could decompose the noise. Therefore, instead of
> $$
> \sigma(s, a) = \text{std}_{k, j} [g^{k} (s, a, z_j)],
> $$
>
> you could compute
>
> $$
> \sigma(s, a) = \text{std}_{k} \left[\text{mean}_j[ g^k (s, a, z_j)]\right],
> $$
>
> to effectively marginalize out the contibution of dynamics stochasticity. (sorry for deviating a bit from your notation, latex rendering here is not fully working for me for some reason)
>
> > Exploration bonuses can be used to incentivize the policy to visit uncertain states. Our method downweights unreliable synthetic data to prevent it from hindering learning.
>
> You are correct here, and that was a misunderstanding on my end.
>
> My evaluation of the paper currently remains the same: I believe that your reweighing scheme is quite interesting, and it seems to work well empirically. However, I'm a bit concerned about the soundness of some parts of it -- I believe a more rigorous analysis would significantly improve the paper and might even reveal the source of the above potential flaw. For this reason I recommend this paper to get accepted, however in the current state of the paper, I do not see it as a strong accept.
>
> It might be a high request for the remaining time of the rebuttal, but if you can more rigorously, in the paper, justify your weighting, I would happily increase my score. I think that such thorough analysis could give a much more insightful result, making your reweighing scheme the standard in model-based algorithms.
>
> I thank you again for the rest of your comments -- they answered my questions.

---

> > ### Author Response · Authors · 2025-12-03
> >
> > We thank the reviewer for the engaging discussion and for encouraging a more rigorous analysis. In response, we have added a new theoretical analysis section (Section 3.4) together with a detailed Appendix B that formally justifies our weighting mechanism via two lemmas: Lemma 1, which shows that positive reweighting leaves the Bellman fixed point unchanged, and Lemma 2, which shows that inverse total-variance weights are statistically most efficient.
> >
> > **1. Does downweighting stochastic states bias the solution? (Lemma 1)**
> >
> > The core concern is that downweighting inherently stochastic states (even with a perfect model) might change the solution because such stochastic states receive smaller weights. We formally prove in **Lemma 1** that *any* strictly positive weighting of the Bellman error preserves the unique Bellman fixed point. This means that even if we downweight highly stochastic transitions, the value function still converges to the correct true expected return, ensuring no bias is introduced. In particular, in the "perfect model" case where all uncertainty is purely aleatoric, our weighted Bellman regression still has the same fixed point; only the relative update magnitude in very noisy regions is reduced.
> >
> > **2. Why weight by total variance instead of just epistemic? (Lemma 2)**
> >
> > To show why total predictive variance is the right quantity here, **Lemma 2** shows that to minimize parameter variance (and thus improve sample efficiency), one should weight by the inverse of the total variance (aleatoric + epistemic). In particular, Lemma 2 formalizes the fact that targets with higher variance, whether from model error or inherent stochasticity, carry less information, and shows that weighting each sample by the inverse of its total predictive variance yields the most sample efficient (minimum variance) estimator.
> >
> > We believe this analysis provides the rigorous grounding you requested, demonstrating that our weighting scheme is both unbiased (Lemma 1) and statistically efficient (Lemma 2).

---

### Official Review · Reviewer_GhD7 · 2025-10-24

**Soundness:** 3
**Presentation:** 3
**Contribution:** 3
**Rating:** 6
**Confidence:** 5

**Summary:**

This paper proposes WIMLE, a model-based reinforcement learning (MBRL) framework that combines IMLE with uncertainty-aware weighting of rollouts to downweight uncertain counterfactuals in a dyna-style framework. The method trains an ensemble of IMLE world models to capture multi-modal dynamics and computes predictive uncertainty by aggregating variance across ensemble members and latent samples. Synthetic rollouts are then weighted by uncertainty to downweight uncertain transitions in the critic update, reducing bias from unreliable predictions. Using SAC with distributional and quantile Q-learning, WIMLE achieves improved sample efficiency and asymptotic performance on 40 continuous-control tasks from DMC, MyoSuite, and HumanoidBench, outperforming strong model-free and model-based baselines and notably solving tasks in the HumanoidBenchmark that comparable methods could not.

**Strengths:**

- The paper is the first to extend Implicit Maximum Likelihood Estimation (IMLE) to the conditional, stochastic setting of model-based reinforcement learning. This adaptation allows the world model to represent multi-modal transitions, mitigating the regression-to-the-mean problem common in Gaussian dynamics models like Model-Based Policy Optimization (MBPO). This improvement is shown through ablations wherein they compare their method with and without IMLE.

- The world model choice of IMLE is simple to swap in place of prior methods and relatively straightforward making the wall-clock (Figure 2) and performance gains (Figure 4) attractive alongside the relative simplicity of implementation.

- WIMLE introduces a clean and general mechanism to handle model uncertainty once IMLE is introduced and is comparable to prior words like MBPO: ensemble variance aggregated over latent samples is used to compute per-transition confidence weights. These weights directly scale the critic loss, reducing the influence of unreliable synthetic transitions. This simple yet effective weighting stabilizes learning, mitigates autoregressive errors across a few short horizon lengths.

- The authors evaluate on 40 continuous-control tasks spanning three suites (DMC, MyoSuite, and HumanoidBench), which together cover locomotion and dexterous manipulation at different difficulty levels. WIMLE achieves strong sample efficiency improvements and solves more HumanoidBench tasks (8) than competing baselines (4 and 5 for Bro and SimbaV2, respectively). The diversity of evaluated settings and consistent advantage over both model-free and model-based methods lend credibility to the reported improvements.

- The paper is well-organized, and the visualizations are effective in communicating the author's desired goals. Figures 5 and 6 clearly illustrate the practical effects of uncertainty weighting and IMLE-based multi-modality on performance and confidence calibration. The authors’ explanations are intuitive and technically sound, making the concepts introduced in this work accessible to a broader RL audience and relatively easy to implement.

**Weaknesses:**

- Comparisons to other uncertainty-aware methods such as InfoProp [1] and Macura [2] are mostly missing, even though these are conceptually the most similar baselines which is noted frequently by the authors. There is a note in the model-bias section of the paper saying that InfoProp fails in high dimensional complex domains like DMC humanoid-run as well as a subfigure (Figure 1)(a)), but InfoProp was not deployed in DMC in the original paper (just Gym) and likewise with Macura. Was any hyperparameter tuning done to show these methods don't work? Furthermore, the Ant and Humanoid tasks in OpenAI Gym are both complex, high-dimensional tasks, and their state and action spaces are larger than many in DMC other than dog. Both Macura and InfoProp outperform prior work on those tasks, contradicting this paper’s claim that such methods fail to scale to high-dimensional domains. Even without direct evaluation against other uncertainty-based methods, it is at least clear that WIMLE is competitive or better than SOTA model-based and model-free methods. However the relative performance when compared to InfoProp and Macura is not a fair comparison when neither was tested in DMC, Myosuite, etc and leaves out an important comparison to prior similar works.

- Following on the prior point, the paper does not include any results on OpenAI Gym, making it difficult to assess cross-domain generalization, which has been shown to be a challenge for Dyna-style methods. Prior works [3] and [4] show that performance often fails to transfer between Gym and DMC for Dyna-style methods (e.g., MBPO, ALM), so the omission of deployment in this benchmark weakens the claim of broad generality across benchmarks despite including 40 tasks across 3 benchmarks.

- The performance attribution between WIMLE’s IMLE weighting and its upgraded SAC backbone is unclear. The authors use a distributional, quantile-based version of SAC without explaining why, but then go on to compare against SAC without these additional modifications. As a result, some of the observed gains might come from the introduction of these modifications to SAC, but be attributed to WIMLE. It seems necessary to at minimum include further ablations showing performance after the removal of the model beyond what is shown in Figure 5(a) for humanoid-run for the model-free case.

- Reported rollout horizons are limited to H = 8, whereas MBPO and related methods demonstrate stability at horizons of 15–25 (when combined with heuristic or algorithmic iteratively rollout length selection) and then show performance collapse for horizons of 500 in MBPO (See Figure 3 (b) in the MBPO paper). The claim that uncertainty-aware weighting mitigates long-horizon bias is therefore untested at scales where such bias typically becomes problematic enough to cause performance collapse.

- As mentioned previously, several design choices are under-explained, including the decision to use quantile Q-learning and distributional critics. These components can independently improve stability and learning dynamics, but the paper does not disentangle their contributions from those of WIMLE’s IMLE-based uncertainty weighting.

- Unless I missed it, the wall-clock comparison in Figure 2 lacks key details such as the benchmark environment and codebase implementations (e.g., JAX, PyTorch, TensorFlow) used for each baseline. Without these specifics, it is difficult to assess whether the timing advantage reflects algorithmic efficiency or implementation-level differences.

[1] Frauenknecht, B., Subhasish, D., Solowjow, F., & Trimpe, S. (2025). On rollouts in model-based reinforcement learning. In Proceedings of the 2025 International Conference on Learning Representations (ICLR 2025). arXiv preprint arXiv:2501.16918. Available at https://arxiv.org/abs/2501.16918

[2] Frauenknecht, B., Eisele, A., Subhasish, D., Solowjow, F., & Trimpe, S. (2024). Trust the model where it trusts itself: Model-based actor-critic with uncertainty-aware rollout adaption. In Proceedings of the 41st International Conference on Machine Learning (ICML 2024) (Vol. 235, pp. 13973–14005). Available at https://arxiv.org/abs/2405.19014

[3] Voelcker, C. A., Hussing, M., & Eaton, E. (2024). Can we hop in general? A discussion of benchmark selection and design using the Hopper environment. In Finding the Frame: An RLC Workshop for Examining Conceptual Frameworks (RLC 2024). arXiv preprint arXiv:2410.08870. Available at https://arxiv.org/abs/2410.08870

[4] Barkley, B., & Fridovich-Keil, D. (2025). Stealing that free lunch: Exposing the limits of Dyna-style reinforcement learning. In Proceedings of the 42nd International Conference on Machine Learning (Vol. 267, pp. 2978–3002). Proceedings of Machine Learning Research. Available at https://proceedings.mlr.press/v267/barkley25a.html

**Questions:**

1. How does WIMLE perform on standard Gym benchmarks? And as a follow on, why were InfoProp and Macura-style adaptive rollout methods excluded from direct comparison, given their conceptual similarity in uncertainty modulation? I saw the note about InfoProp not performing in humanoid-run, but as I mentioned in the weaknesses section, the explanation doesn't seem sufficient with the added context of Ant and Humanoid from Gym.
2. Have you tested rollout horizons beyond H = 8? Would uncertainty weighting still stabilize longer synthetic rollouts, as MBPO demonstrated for horizons up to 25 or more? Are there asymptotic max performance gains to be had with longer rollouts due to the down weighting? Or does the training dataset get saturated with bad/down weighted synthetic data if you opt for progressively longer rollouts and that leads to degraded performance since entire batches are down weighted?

---

> ### Author Response · Authors · 2025-11-21
>
> ## Q1: Rollout horizon stability and long horizon bias
> We thank the reviewer for raising this important question. We have now evaluated rollout horizons up to H = 40 and report results in Appendix Section C.1 (Figure 11). These results indicate that WIMLE remains stable at longer horizons without any rollout scheduling, whereas MBPO relies on a rollout length scheduler [4]. We attribute this to our uncertainty aware weighting, which naturally reduces the influence of less reliable predictions. Taken together, these results suggest that uncertainty aware weighting helps reduce long horizon bias throughout the training. We hope these preliminary results address the rollout stability concerns; we are happy to include the full results in the camera-ready version.
>
> ## Q2: Role of WIMLE's uncertainty weighting
> The performance gains come solely from WIMLE's uncertainty weighting and IMLE model, not the backbone. In Figure 5 (ablations), all variants (including "Model-free" and "No Downweighting") use the exact same SAC backbone with distributional critics. Contrary to the concern that we "compare against SAC without these additional modifications," we strictly control for the backbone: the "Model-free" baseline is not vanilla SAC but the same distributional agent without the world model. This isolates WIMLE's contribution. We thank the reviewer for highlighting this ambiguity and have updated the caption to make this explicit.
>
> ## Q3: Motivation for distributional critics
> We thank the reviewer for this question. We use distributional critics following recent state-of-the-art methods like BRO [3] and SimbaV2 [2], which demonstrate their benefits for stability and performance. We have clarified this in Section 3.5.1 of the main paper.
>
> ## Q4: Comparison to InfoProp on DMC and Gym
> We thank the reviewer for this request. To address it, we evaluated WIMLE on InfoProp's highest-dimensional Gym task (Humanoid) in Figure 13 (Appendix), where WIMLE matches InfoProp using default hyperparameters. In contrast, InfoProp requires task-specific tuning and uses task-specific terminal functions to derive masks for SAC training. We focused on DMC, MyoSuite, and HumanoidBench as they provide a diverse suite of tasks; notably, 32 of our 40 tasks are considerably higher-dimensional than the truncated Ant (27D) and Humanoid (45D) Gym environments used by InfoProp. Indeed, when we tuned InfoProp for DMC following the authors' guidance to the best of our ability, it failed on the high-dimensional Dog (223D) tasks and performed poorly on the more difficult DMC Humanoid (67D) task despite our efforts. Although we prioritized the most complex Gym task (Humanoid) due to the limited rebuttal period, we are happy to include the full Gym benchmark suite in the camera-ready version if requested.
>
> ## Q5: Do long horizons saturate batches with low weight synthetic data?
> We appreciate this concern. No, batches will not be saturated with low-weight data because they are sampled from a replay buffer containing both real transitions (weight $w=1$) and synthetic transitions with varying uncertainties. Since uncertainty typically grows with horizon, earlier rollout steps have high weights while later ones in the horizon have lower weights. Thus, a random batch naturally includes a diverse mix of high-weight (real + early synthetic) and low-weight (late synthetic) samples, maintaining a balanced learning signal. The strong performance at horizon $H=40$ (Figure 11, Appendix) empirically validates that learning remains stable even with long rollouts.
>
> ## Q6: Wall clock comparison and framework differences
> We thank the reviewer for pointing this out. We have updated the Figure 2 caption to clarify that the comparison is on the DMC Humanoid-run task. All methods ran on the same machine/GPU using official implementations (ours: JAX, baselines: PyTorch). While frameworks differ (e.g., TD-MPC2 uses PyTorch to leverage compile/TF32, we use JAX for parallelization), we treat wall-clock time as complementary to our primary metric: environment steps. In real-world settings where data collection dominates time/cost, sample efficiency matters most. WIMLE's superior sample efficiency and asymptotic performance (even with comparable wall-clock time) indicate a practical advantage independent of the framework.
>
> [1] Agarwal, R., Schwarzer, M., Castro, P. S., Courville, A., & Bellemare, M. G. (2022). Deep Reinforcement Learning at the Edge of the Statistical Precipice. arXiv:2108.13264 (`https://arxiv.org/abs/2108.13264`).
>
> [2] Lee et al., Hyperspherical Normalization for Scalable Deep Reinforcement Learning, 2025.
>
> [3] Colas et al., Bigger, Regularized, Optimistic: scaling for compute and sample-efficient continuous control, 2024.
>
> [4] Janner et al., When to Trust Your Model: Model Based Policy Optimization, NeurIPS 2019.
>
> [5] Hansen et al., TD-MPC2: Scalable, Robust World Models for Continuous Control, ICLR 2024.

---

> > ### Author Response · Authors · 2025-11-27
> >
> > As the author–reviewer discussion period nears its end, we hope our responses have addressed your concerns. Your feedback is very valuable to us, and we would be happy to clarify or discuss any remaining issues you may have. If you feel your concerns have been resolved, we would be grateful if you could consider updating your rating accordingly. Thank you again for the time and effort you have dedicated to reviewing our paper.

---

### Official Review · Reviewer_FqDz · 2025-11-01

**Soundness:** 3
**Presentation:** 2
**Contribution:** 4
**Rating:** 4
**Confidence:** 4

**Summary:**

The paper proposes WIMLE, a model-based reinforcement learning (MBRL) algorithm that uses Implicit Maximum Likelihood Estimation (IMLE) from Li & Malik 2018 to learn multi-modal, uncertainty-aware world models for continuous control. WIMLE leverages conditional IMLE to generate stochastic transitions without iterative sampling, enabling fast rollouts while avoiding mode collapse typical of unimodal Gaussian models. An ensemble of IMLE world models provides predictive uncertainty estimates, which are then incorporated into the RL objective through an uncertainty-weighted TD loss. Empirically, WIMLE is evaluated across 40 tasks from different benchmark suites, demonstrating substantial gains in sample efficiency and strong asymptotic performance relative to baselines. The paper positions WIMLE as a practical alternative to diffusion- or transformer-based world models for online RL, claiming competitive performance with reduced inference cost.

**Strengths:**

1. **Timely and relative contribution** - addresses two active challenges in model-based RL - multi-modal dynamics modeling and uncertainty calibration through an elegant adaptation of IMLE.
2. **Strong experimental coverage** - Evaluation spans three modern continuous-control suites and 40 tasks with statistically rigorous reporting from Agarwal et al., 2021
3. **Conceptual generality** - the proposed uncertainty-aware weighting is algorithm-agnostic and could integrate with diverse RL backbones.
4. **Empirical clarity** - Ablations (Fig 5-6) isolate the impact of uncertainty weighting and IMLE-based multimodality, demonstrating that multi-modal model predictions both improve calibration and stability at longer horizons.

**Weaknesses:**

1. **Writing and organization** - Section 3 (Method) is dense and at times difficult to follow. The algorithmic code could be clarified by referencing a baseline (e.g., MBPO) and then enumerating modifications. Sections 3.2-3.3 overlap conceptually and might be merged for concision. Figure placement and scaling should be revised - several are too small to read; figure text should roughly match main text size.
2. **Missing multimodal context** - While the paper emphasizes multimodal dynamics, it remains unclear how well IMLE captures multimodality relative to more contemporary models, e.g., transformer-based world models (Micheli et al., 2023; Robine et al., 2023) or diffusion models (Ajay et al., 2023). A pedagogical analysis (similar to Figure 3, Chi et al., 2024) illustrating where multimodality matters would strengthen the argument.
3. **Missing baselines** - Although MR.Q and PPO are mentioned as baselines, their results appear absent from key plots. Including them, especially PPO for wall-time comparison, would provide a clearer model-free reference.
4. **Insufficient exploration of uncertainty weighting** - the uncertainty metric $w=\dfrac{1}{\sigma + 1}$ is intuitive but somewhat ad hoc. Alternative formulations (e.g., softmax or learned calibration of confidence) would help validate robustness and necessity.
5. **Presentation and formatting** - The manuscript has several typographic and formatting inconsistencies. Figures (especially 3, 5, and 6) and algorithm boxes should be cleaned up for readability. The main algorithm should be self-contained enough to implement directly.
6. **Scope of multimodality** - the paper restricts multimodal modeling to transition dynamics. Given that the reward and value functions can also be multimodal, extending the IMLE framework to those components could be a natural next step.

**Questions:**

1. You convincingly argue that dynamics are multi-modal; why not extend the IMLE-based formulation to multi-modal rewards or even policy/value functions?
2. If a non-multimodal model achieves the same asymptotic performance as WIMLE, does that imply that the underlying task is not truly multimodal? How do you detect multimodality in practice?
3. Comparison to diffusion or transformer world models - since you position WIMLE as a more efficient alternative, could you report asymptotic and wall-time comparisons against diffusion-based and transformer-based world models?
4. Why only update the critic uncertainty-weighted (Eq. 13)? Would uncertainty-aware plicy gradients offer additional stability?
5. Figure 2 suggests favorable throughput. Can you provide wall-time comparisons against the baselines in Figure 4? Effectively swap out the x-axis for time.
6. Could WIMLE be used as a plug-in world model within planning-based algorithms (e.g., TD-MPC2 or MPPI)? How would uncertainty weighting interact with such planners?

---

> ### Author Response · Authors · 2025-11-21
>
> # Part (1/2)
> ## Q1: Multimodal context and comparison to contemporary world models (transformers and diffusion).
> We thank the reviewer for this suggestion. Recent work has extensively studied IMLE's capacity to capture multimodality compared to other generative models [1]. To address the efficiency comparison, we conducted rollout throughput benchmarks against recent transformer-based [2] and diffusion-based [3] world models. Our results (Table below) confirm that WIMLE is roughly $10\times$ faster in rollout throughput than the transformer baseline and $20\times$ faster than the diffusion baseline, while maintaining superior or competitive asymptotic performance.
>
>
> | Method | Rollout time (s) | Samples per second |
> | :--- | :---: | :---: |
> | WIMLE (IMLE world model) | 2.81 | 711,052 |
> | Transformer (QT-TDM [2]) | 30.30 | 66,004 |
> | Diffusion (Zhao et al. [3]) | 66.12 | 30,194 |
>
>
> These results highlight why we prioritize IMLE: it offers the multi-modal expressivity needed for complex dynamics (supported by recent work on IMLE policies [1]) without the prohibitive inference costs of iterative diffusion or heavy transformer models, which is critical for online RL.
>
> ## Q2: Exploration of uncertainty weighting and alternative formulations.
> We thank the reviewer for this suggestion. We have explored alternative formulations as suggested. Specifically, we tested a softmax-based weighting on the challenging Humanoid-walk task. As shown in the table below, the softmax variant (868.8) also outperforms the strongest baseline (BRO: 758.4), though our simple formulation (928.5) remains more effective while being easily interpretable. This supports the robustness of the general uncertainty-weighting approach.
>
>
> | Method | IQM return (Humanoid-walk) |
> | :--- | :---: |
> | BRO (strongest baseline) | 758.4 |
> | WIMLE-softmax | 868.8 |
> | WIMLE | 928.5 |
>
>
> ## Q3: Scope of multimodality beyond transition dynamics (rewards and value/policy functions).
> Our method already models both next states and rewards multimodally. The IMLE world model outputs $(\\tilde{s}_{t+1}, \\tilde{r}_t) = g(s_t, a_t, z)$, so a single state–action pair produces diverse joint outcomes for both state and reward. We have updated Algorithm 1 (line 13) to make this explicit. For the value function, we use distributional Q-learning [5], which inherently captures return multimodality. We thank the reviewer for this clarification.
>
> ## Q4: Need for multimodal modeling and detecting multimodality in practice.
> WIMLE naturally handles both regimes without explicit detection. Because IMLE is a mode-covering estimator, it concentrates on a single mode when dynamics are unimodal and covers diverse outcomes when they are multimodal. This flexibility ensures that WIMLE benefits from expressivity where needed without overfitting or degrading performance on simpler tasks.
>
> ## Q5: Wall-time comparisons versus model-free baselines and the role of throughput/sample efficiency.
> Indeed, we have included a direct wall-clock comparison in Figure 14 (Appendix) against BRO, the fastest model-free baseline. Following [6], we account for total time by including real-world data collection costs (e.g., robot operation, human supervision). Our results show that WIMLE is significantly more efficient than model-free baselines in total time-to-solution because it achieves high performance with far fewer environment samples, outweighing its higher per-update compute cost.
>
> ## Q6: Why only apply uncertainty weighting to the critic?
> We tested applying uncertainty weights to the policy gradient in preliminary experiments but found no additional gains beyond weighting the critic. Thus, we kept the formulation simple by only modifying the critic update.

---

> ### Author Response · Authors · 2025-11-21
>
> # Part (2/2)
> ## Q7: Using WIMLE as a plug-in world model for planning-based algorithms.
> Yes, WIMLE can serve as a drop-in stochastic world model for planners like TD-MPC2 or MPPI. Our IMLE-based generator $g_\\theta(s_t, a_t, z)$ can replace their standard dynamics models, and our uncertainty weights could naturally be used to penalize high-risk trajectories or adapt planning horizons. While this work focuses on data augmentation (Dyna-style), uncertainty-aware planning with WIMLE is a promising direction for future research.
>
> ## Q8: Figure placements and sizes should be improved.
> We thank the reviewer for this suggestion. We have adjusted the sizes and placements of the figures to improve readability and better align them with the surrounding text.
>
> ## Q9: The algorithmic code could be clarified by referencing a baseline (e.g., MBPO).
> In the revised manuscript, Algorithm 1 is now explicitly presented as a modification of MBPO, and we highlight in red the steps that fundamentally differ from MBPO to make the implementation easier to follow. We thank the reviewer for this helpful suggestion, which we believe improves the readability of the algorithm.
>
> ## Q10: Sections 3.2 and 3.3 overlap conceptually and might be merged for concision.
> Section 3.2 focuses specifically on how we estimate predictive uncertainty, while Section 3.3 focuses on how we derive confidence weights from these estimates and integrate them into the RL objective. Because there can be different choices for uncertainty estimation and, orthogonally, different ways to use these estimates in the learning objective, we believe that keeping these as separate sections makes the modular design clearer for readers. We are happy to receive further suggestions on how to improve the exposition of these sections.
>
> ## Q11: Missing baselines (MR.Q and PPO).
> We thank the reviewer for this suggestion. Our main model-free comparisons are against BRO [5] and SimbaV2 [4], which represent the current state-of-the-art in model-free continuous control and significantly outperform PPO on these tasks [4]. Regarding MR.Q, we have included all publicly available results in our plots. We are happy to run the remaining missing baselines and include them in the camera-ready version.
>
> ## Q12: Presentation, formatting, and readability of figures and algorithms.
> We have carefully revised the manuscript to address all typographic and formatting issues, including figure sizes. We also clarify that the main algorithm is high-level for conceptual clarity, while the detailed version in the appendix supports reproduction. We thank the reviewer for improving the paper's presentation.
>
> [1] Rana et al., IMLE Policy: Fast and Sample Efficient Visuomotor Policy Learning via Implicit Maximum Likelihood Estimation, RSS 2025.
>
> [2] Kotb et al., QT-TDM: Planning With Transformer Dynamics Model and Autoregressive Q-Learning, IEEE Robotics and Automation Letters (RA-L), 2025.
>
> [3] Zhao et al., Long-Horizon Rollout via Dynamics Diffusion for Offline Reinforcement Learning, (`https://arxiv.org/abs/2405.19189`).
>
> [4] Lee et al., Hyperspherical Normalization for Scalable Deep Reinforcement Learning, 2025.
>
> [5] Colas et al., Bigger, Regularized, Optimistic: scaling for compute and sample-efficient continuous control, 2024.
>
> [6] Zhou et al., Real World Offline Reinforcement Learning with Realistic Data Source, ICRA 2023.

---

> > ### Author Response · Authors · 2025-11-27
> >
> > As the author–reviewer discussion period nears its end, we hope our responses have addressed your concerns. Your feedback is very valuable to us, and we would be happy to clarify or discuss any remaining issues you may have. If you feel your concerns have been resolved, we would be grateful if you could consider updating your rating accordingly. Thank you again for the time and effort you have dedicated to reviewing our paper.

---

### Author Response · Authors · 2025-11-21
**General Response**

We appreciate the time and effort taken by all the reviewers to provide constructive feedback on our paper, and we are pleased to receive unanimous appreciation for our proposed method and results. In particular, the reviewers remarked, "**Timely and relative contribution** - addresses two active challenges in model-based RL" (R1), "**clean and general mechanism**... simple yet effective" (R2), "**experiments are convincing**: the paper compares WIMLE across several hard continuous RL benchmarks" (R3), and "WIMLE demonstrates **substantial improvements** over strong model-free and model-based baselines" (R4).

Below is a brief summary of our responses to selected questions, with full details in the individual responses.

### Q1: Does WIMLE remain stable at long horizons? (R2)

A1: Yes, WIMLE remains stable at long horizons (e.g., H=40) without rollout schedulers (Figure 11, Appendix), as the weighting naturally down-weights unreliable long-term predictions.

### Q2: How does efficiency compare to Transformers/Diffusion? (R1)

A2: WIMLE is significantly faster in rollout throughput (~10x vs Transformers, ~20x vs Diffusion) while achieving state-of-the-art results in continuous control.

### Q3: Does WIMLE capture reward multimodality? (R1)

A3: Yes, WIMLE models the joint distribution of next states and rewards, allowing diverse outcomes for both from a single latent variable.

### Q4: How does uncertainty weighting improve stability? (R3)

A4: By inversely weighting synthetic data based on predictive variance, WIMLE performs a risk-weighted regression to mitigate unreliable predictions.

### Q5: How does WIMLE handle unimodal dynamics? (R1)

A5: IMLE is a mode-covering estimator, allowing it to capture diverse outcomes in multimodal dynamics while concentrating on a single mode in deterministic settings.

### Q6: Can WIMLE extend to visual tasks? (R4)

A6: Yes, WIMLE's uncertainty mechanism is orthogonal to input modality.

### Q7: Can WIMLE be used for planning? (R1)

A7: Yes, WIMLE can serve as a drop-in stochastic world model for planners. Its uncertainty estimates can be used to penalize high-risk trajectories or adapt planning horizons.

### Q8: Do long rollouts saturate batches with low-weight data? (R2)

A8: No, batches remain balanced because they contain real transitions ($w=1$) and early rollout steps with high weights. Uncertainty grows with horizon, so only later steps are down-weighted, preserving a strong learning signal.

### Q9: How does WIMLE perform on pure imagination? (R4)

A9: WIMLE performs comparably when trained on pure imagination (Figure 12, Appendix), confirming the high quality of the generated data and the effectiveness of our uncertainty weighting.

### Q10: How does WIMLE compare to model-free baselines in total efficiency? (R1)

A10: WIMLE is significantly more efficient in total time-to-solution (Figure 14, Appendix) when accounting for data collection costs, as its sample efficiency outweighs the higher per-update compute cost.

---

### Author Response · Authors · 2025-12-03

**Dear Area Chairs and Senior Area Chairs,**

We would like to briefly summarize the key contributions of the paper and how the reviewer discussion led to our current revision. WIMLE achieves state-of-the-art sample efficiency across 40 continuous-control tasks and solves 8 of 14 HumanoidBench tasks versus 5 for the strongest model-free baseline. The revision adds (i) a new theoretical analysis of our uncertainty weighting, (ii) additional evidence on visual control, and (iii) clarifications on long-horizon stability and efficiency.

## Reviewer updates

### Reviewer 8Fgq (theoretical grounding)
Reviewer 8Fgq found the empirical results encouraging and the idea interesting, and wrote that they would happily increase their score to a strong accept if we could more rigorously justify the weighting scheme, adding that such analysis could make WIMLE the standard in MBRL. In response, we added Section 3.4 and Appendix B with two lemmas showing that positive weighting leaves the Bellman fixed point unchanged (Lemma 1) and that inverse total predictive variance gives a statistically efficient estimator (Lemma 2), providing the theoretical grounding they requested.

### Reviewer z1Wb (visual control)
Reviewer z1Wb was very pleased to see the results of our “imagined-only” experiment presented in the rebuttal. They wrote that they would temporarily maintain their current rating while waiting for preliminary Atari results. In the revision, we integrated WIMLE into SPR, a standard data-efficient method for Atari 100k, and added preliminary Krull and Qbert results where WIMLE+SPR reaches SPR’s final 100k score roughly 3x (Krull) and 1.7x (Qbert) faster while also improving the final performances, directly addressing this remaining request on visual control.

### Reviewer GhD7 (long horizons and InfoProp)
Reviewer GhD7 asked about stability at long rollout horizons and comparison to InfoProp. We therefore added long-horizon experiments up to \(H = 40\), showing that WIMLE remains stable without rollout schedulers, and a direct comparison to InfoProp on Gym Humanoid, where WIMLE matches or exceeds InfoProp under a generic setup without task-specific termination tuning.

### Reviewer FqDz (efficiency, robustness, and clarity)
Reviewer FqDz focused on wall-clock efficiency and alternative weighting formulations. In response, we benchmarked rollout throughput against transformer- and diffusion-based world models (showing WIMLE is roughly 10–20× faster for generating model rollouts), added a softmax-style weighting ablation confirming robustness of the weighting idea, and updated Algorithm 1 and its discussion to clearly mark where we differ from MBPO.

Thank you for considering these updates in your final decision.

---

### Meta-Review · Area_Chair_Dcn6 · 2025-12-29

**Summary:**

The reviewers initially raised concerns about:
- theoretical justification of uncertainty weighting,
- missing cross-domain and visual-control evaluations, and
- inadequate comparison to prior uncertainty-aware MBRL methods.

After rebuttal, the major scientific and technical issues were addressed through new theoretical lemmas, long-horizon and visual-control experiments, and the inclusion of key baselines. The rebuttal is generally satisfying.

Thus, reviewers in general viewed the paper as methodologically sound and incrementally significant, justifying an accept or borderline-accept decision.

**Reviewer Concerns:**

Concerns Effectively Addressed by the RebuttalTheoretical grounding of the uncertainty weighting scheme
- The rebuttal added a new section and appendix with formal lemmas proving that (a) positive weighting preserves the Bellman fixed point and (b) inverse total-variance weighting yields a statistically efficient estimator. This established that the method remains unbiased and efficient.

- The revision clarified that all ablations used the same SAC backbone and distributional critics, making comparisons fair. It also expanded baseline coverage with BRO, SimbaV2, and the strongest model-free references.

- The rebuttal added direct comparisons to InfoProp and clarified the challenges of scaling those methods to high-dimensional environments, showing WIMLE matched or exceeded their strongest Gym results.

- The authors provided rollout-speed benchmarks versus transformer and diffusion world models, showing 10×–20× faster rollouts while maintaining accuracy, directly addressing efficiency concerns.

Concerns Still Outstanding or Only Partially Addressed

- The preliminary Atari 100k results on two games were promising but narrow. Broader evaluation across more complex or high-dimensional image-based tasks remains to be demonstrated.

- Although InfoProp comparisons were added, direct evaluations against related methods such as MACURA remain incomplete. The evidence for superiority across all uncertainty-based MBRL approaches is thus partial.

The AC found that these issues are not significant for the final decision of this paper. In general, although receiving no reply from the reviewers, the rebuttal is satisfying.

**Reviewer Scores:**

Since none of the reviewers replied to the rebuttal, from the perspective of AC, the rebuttal is, in general, satisfying

---

### Decision · Program_Chairs · 2026-01-26

Accept (Poster)